

# Boundary signature of singularity in the presence of a shock wave

Gary T. Horowitz[1], Henry Leung[1], Leonel Queimada[1] and Ying Zhao[2]

**1** Department of Physics, University of California, Santa Barbara, CA 93106
**2** Kavli Institute for Theoretical Physics, Santa Barbara, CA 93106

## Abstract

Matter falling into a Schwarzschild-AdS black hole from the left causes increased focussing of ingoing geodesics from the right, and, as a consequence, they reach the singularity sooner. In a standard Penrose diagram, the singularity "bends down". We show how to detect this feature of the singularity holographically, using a boundary two-point function. We model the matter with a shock wave, and show that this bending down of the singularity can be read off from a novel analytic continuation of the boundary two-point function. Along the way, we obtain a generalization of the recently proposed thermal product formula for two-point correlators.



# 1   Introduction

AdS/CFT duality [1–3] gives us a concrete framework and powerful tools to study quantum gravity. There has been much progress since it was originally proposed more than 25 years ago. In particular, significant progress has been made in understanding black holes, thanks to the ideas from quantum information, quantum chaos, etc. There is a close connection between the near-horizon region and quantum chaos. The behavior of infalling objects near the horizon, like the exponential growth of momentum, back-reaction on the geometry due to one or two colliding objects, etc, can all be understood through concepts like operator growth, out-of-time-ordered correlators, and a quantum circuit model of the dual quantum mechanical theory [4–8].

However, despite extensive research and rapid progress, one aspect of the black hole remains mysterious, and that is the central singularity. The singularity is the place where space-time ends and classical general relativity breaks down. How do we understand this from the dual quantum mechanical theory? What is the quantum information meaning of the singularity? In fact, it is not even clear what is the right concrete question to ask about the singularity. It is likely that a better understanding of the black hole singularity will also lead to a better understanding of the big bang singularity.

Several attempts to use holography in order to understand the black hole singularity have been made [9–14]. In [9] the authors found subtle signatures of the curvature singularity in certain analytically continued boundary two-point functions for Schwarzchild-AdS black holes. The authors of [10] made it more transparent and pointed out that the signature is encoded in frequency space two-point functions where the frequency is taken to be imaginary and large. In [12] the authors read out the proper time from the bifurcation surface to the singularity using thermal one-point functions. All of the above computations were done in the context of the unperturbed Schwarzchild-AdS background and relied on the analytic properties of that spacetime.[1] More general spacetimes containing black holes are not analytic.

In this paper, we will study perhaps the simplest example of a nonanalytic black hole, i.e., Schwarzschild-AdS with a shock wave at the horizon. This spacetime is dual to a thermofield double state with a perturbation at time $t_w$ in the limit that $t_w \to -\infty$. Matter falling into the black hole from the left boundary causes increased focusing of ingoing geodesics from the right, and, therefore, they reach the singularity sooner. In a standard Penrose diagram, the singularity "bends down". The amount of focusing will depend on the time when the ingoing geodesics leave the right boundary. Our goal is to look for boundary signals of such "bending down" behavior of the singularity.

To be more concrete, suppose we send in a signal from the right boundary at time $t_R$, and let $t_L(t_R)$ be the latest left boundary time at which someone who jumps in can receive the signal.[2] Consider the quantity $t_R + t_L(t_R)$. Without any perturbation, the boost symmetry

---

[1]See also [15–20] for recent works related to these approaches.

[2]Throughout this paper we assume the bulk time $t$ increases toward the future in the right exterior, and increases

ensures that this is a constant, independent of $t_R$. It vanishes for a BTZ black hole, which reflects the fact that the Penrose diagram is a square. On the other hand, $t_R + t_L(t_R) < 0$ for higher dimensional Schwarzchild-AdS black holes, which reflects the fact that, in the standard way of drawing Penrose diagrams, the singularity is no longer represented by a horizontal line, instead it bends down (Figure 1). With a shock wave coming in from the left boundary, we will see that $t_L(t_R)$ is decreased (in a $t_R$ dependent way), which reflects the fact that positive energy matter causes the singularity to bend down further.

Our approach will combine elements of both [9] and [10]. Ref. [9] studied boundary two-point functions and analytically continued in the time variable. Ref. [10] studied frequency space two-point functions and analytically continued to imaginary frequency. Since the shock wave breaks the usual boost symmetry, the correlation function we study will depend on two variables. By working with one time and one frequency, the correlation function has the following key feature: as we take the frequency large and imaginary, while keeping a real boundary time, the dominating contribution comes from real spacetime geodesics and directly encodes the quantity $t_L(t_R)$ defined above.

It has recently been shown [21] that the thermal two-sided two-point correlator can be expressed as a product over quasinormal mode frequencies. In the course of our analysis, we obtain an extension of this result. In the presence of a shock wave at the horizon, the two-sided two-point correlator can be expressed as a product over quasinormal modes and Matsubara frequencies.

The rest of the paper is organized as follows. In Section 2 we give some necessary background and briefly review the results in [9,10]. We also compute $t_L(t_R)$ for the Schwarzschild-AdS black hole with a shock wave, and analyze the "bending down" behavior using the classical geometry. As the calculation of the two-point correlator is somewhat technical, in Section 3 we give an outline of it including the main results, before giving the detailed calculations in Section 4. In Section 5 we point out unanswered questions and future directions. The appendices contain some additional details. The generalization of the thermal product formula is presented in Appendix B.

## 2 Background and motivation

### 2.1 Review: Signature of singularity in analytically continued two-point functions

In this subsection, we briefly review the results in [9] and [10]. In [9], the authors considered an eternal Schwarzchild-AdS black hole in dimension $D > 3$. Such black holes have a spacelike curvature singularity in their interior. Furthermore, boundary anchored spacelike geodesics can get arbitrarily close to the singularity. Such geodesics become almost null as their turning point approaches the singularity and were called bouncing geodesics (see Figure 1).

In holography, boundary correlators can be computed by taking an appropriate rescaled limit of bulk correlators as they approach the boundary. Consider a left-right correlator of a massive bulk scalar field $\phi$ at two points in the asymptotic region at boundary times $t_L$ and $t_R$: $G(t_L, t_R) \equiv \langle \phi(t_L)\phi(t_R) \rangle$. In a Hartle-Hawking state, the boost symmetry implies that the correlator only depends on the combination of the two times $t = t_R + t_L$. Under certain circumstances [22], when the field has large mass, a saddle point approximation relates such correlator to the length of geodesics connecting the two points: $G(t_L, t_R) \sim e^{-mL}$. The authors of [9] tried to exploit such a relation to look for a boundary signature of the singularity. However they found that such bouncing geodesics actually do *not* dominate the correlator.

---

toward the past in the left exterior. However, the boundary times $t_{R/L}$ always increase toward the future.

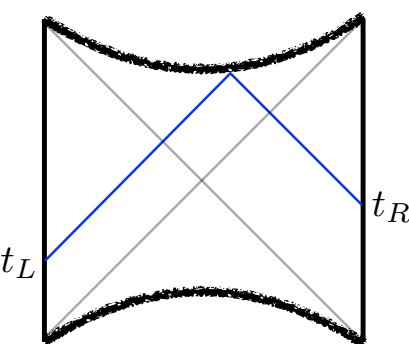

Figure 1: Boundary anchored spacelike geodesics can "bounce" close to the singularity and approach the null geodesics shown.

What dominates the correlator is the sum of contributions from two complex geodesics. To proceed, the authors considered the correlator as an analytic function of boundary time $t$ and continued to a different sheet, where they found a "lightcone singularity" in the correlator which is a subtle signature of the singularity in the interior.

In [10], the authors further studied this question. Instead of doing an analytic continuation in $t$, the authors studied the frequency space correlator

$$G(\omega) \equiv \int_{-\infty}^{+\infty} dt\, e^{i\omega t} \left\langle \mathcal{O}\left(-i\frac{\beta}{2}\right)\mathcal{O}(t)\right\rangle. \tag{1}$$

They found that at large imaginary frequency $\omega = -iE$, the behavior of the correlator is captured by the bouncing geodesic. It has exponential decay $G(\omega) \sim e^{-E\tilde{\beta}/2}$ where $\frac{\tilde{\beta}}{2} = -(t_L + t_R)$, and $t_{R/L}$ is the boundary time at which the bouncing geodesic intersects the boundary (see Figure 1). As discussed in the introduction, this specific combination $t_R + t_L$ is a measure of how much the singularity bends down.[3]

Note that in both papers [9] and [10], the authors assumed that the boundary state is the thermofield double. This is a very special state, which has a boost symmetry and, as a result, the correlator is a function of only one time $t$, or one frequency $\omega$. In this paper, we will consider a more general situation. In particular, we will study the thermofield double state with an early time perturbation, which is dual to a Schwarzschild-AdS black hole with a shock wave present.

## 2.2  Bending down of singularity in the presence of a shock wave

It is well known that a thermofield double state in the field theory is dual to a Schwarzschild-AdS black hole [23]. Its metric is given by

$$ds^2 = -f(r)dt^2 + \frac{dr^2}{f(r)} + r^2 d\Omega_{D-2}^2, \tag{2}$$

where[4]

$$f(r) = r^2 - \frac{\mu}{r^{D-3}} + 1. \tag{3}$$

---

[3]Note that we are using a slightly different convention than [10]. We assume $t$ is the boost-invariant combination $t_R + t_L$ while they assumed $t = -(t_R + t_L)$.

[4]We set the AdS length to be $L = 1$.

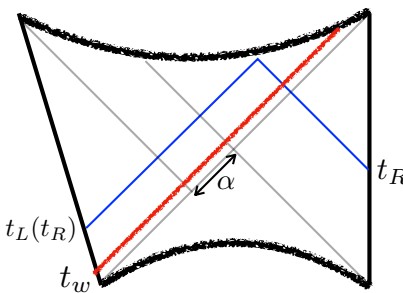

Figure 2: An early perturbation causes the singularity to bend down further, decreasing $t_L(t_R)$.

The horizon is at $r = r_0$ where $f(r_0) = 0$, and the inverse temperature is $\beta = 4\pi/f'(r_0)$. It is convenient to define Kruskal coordinates. Define the tortoise coordinate

$$r_* = -\int_r^\infty \frac{dr}{f(r)}, \tag{4}$$

and introduce null coordinates $u = t - r_*$, $v = t + r_*$. Then the Kruskal coordinates are defined as $U = -e^{-2\pi u/\beta}$, $V = e^{2\pi v/\beta}$.[5]

The quantity introduced above, $\tilde{\beta}$, can be computed from the metric via

$$\int_0^\infty \frac{dr}{f(r)} = \frac{1}{4}(\tilde{\beta} \pm i\beta), \tag{5}$$

where the imaginary part arises by going around the pole at $r_0$ and the sign is determined by the contour chosen. The expression on the left is just the Schwarzschild time difference between the singularity and boundary along a radial null geodesic.

The bulk geometry dual to the thermofield double state with an early time perturbation was studied in [4]. Suppose a perturbation with energy $E$ comes in from the left boundary at time $t_w$. We consider the case where $E$ is very small, and $-t_w$ is very large. The resulting geometry has a shock wave lying near the horizon. As shown in [4], this solution consists of two copies of Schwarzschild-AdS glued together along the shock wave with a shift in the Kruskal $V$ coordinate by $\alpha \sim \frac{E}{M} e^{-\frac{2\pi}{\beta} t_w}$ (see Figure 2).

Physically, as a result of gravitational focusing, the time an infalling observer can experience after crossing the horizon will decrease. Intuitively, one can say the singularity bends down in the Penrose diagram.

To be quantitative, we can consider a beam of radial light rays with certain energy coming in from the right boundary at time $t_R$. Suppose, without the shock wave, there is affine distance $\lambda_0$ between the light rays crossing the horizon and hitting the singularity. Clearly $\lambda_0$ does not depend on $t_R$. One can ask: With the shock wave present, what is the new affine distance experienced by the light rays? The expansion is given by

$$\theta = \frac{1}{A}\frac{dA}{d\lambda} = \frac{(D-2)}{r}\frac{dr}{d\lambda}. \tag{6}$$

Since $r$ is a function of $UV$, and $V$ shifts across the shock, we can write[6]

$$r = h[U(V + \alpha\Theta(U))], \tag{7}$$

---

[5]In the left exterior we let $t \to t - i\frac{\beta}{2}$.

[6]In these $U, V$ coordinates, the metric is continuous across the shock wave. If one defines $\tilde{V} = V + \alpha\Theta(U)$, the metric picks up a $\delta(U)dU^2$ term and is not continuous.

for some function $h$. Assuming the shock lies on the horizon, $U = 0$, we have $\frac{dr}{d\lambda} = h'(0)[V + \alpha\Theta(U)]\frac{dU}{d\lambda}$, which implies

$$\theta = (D-2)\frac{h'(0)}{h(0)}[V + \alpha\Theta(U)]\frac{dU}{d\lambda}. \tag{8}$$

This shows that $\theta$ jumps across the shock wave by an amount

$$\delta\theta = (D-2)\alpha\frac{h'(0)}{h(0)}\frac{dU}{d\lambda}. \tag{9}$$

From (8) we see that the expansion of the light rays right after passing the shock wave, $\tilde{\theta}$, is related to $\theta$ just before by

$$\tilde{\theta} = \theta\left(1 + \alpha V^{-1}\right). \tag{10}$$

Note that both $\theta$ and $\tilde{\theta}$ are negative since $r$ is decreasing.

The Raychaudhuri equation now implies that the affine distance to the singularity is

$$\tilde{\lambda}_0 = \frac{D-2}{\tilde{\theta}} = \frac{D-2}{\theta}\frac{1}{1 + \alpha V^{-1}} = \frac{\lambda_0}{1 + \alpha e^{-\frac{2\pi}{\beta}t_R}} < \lambda_0. \tag{11}$$

We see that this affine distance does depend on $t_R$. It approaches $\lambda_0$ when $t_R$ is large, since the shift in $V$ becomes negligible compared to $V$ when $V$ is large. Physically, the energy of the shock seen by the light ray goes to zero. But the affine distance becomes shorter and shorter as we take $t_R$ earlier, as the shock wave very close to the horizon will have a significant effect on the infalling light ray. This fact is one manifestation of the statement that the singularity bends down.

Another way to compute the jump in $\theta$ across the shock wave (9) is to include the stress tensor of the shock

$$T_{UU} = -\frac{1}{8\pi G_N}(D-2)\alpha\delta(U)\frac{h'(0)}{h(0)}, \tag{12}$$

in the Raychaudhuri equation.

In this paper, we will study another quantity which can characterize this effect. We again consider sending in a signal from the right boundary at time $t_R$, and ask: What is the latest time someone can jump in from the left boundary and still receive the signal? As in Section 1, we call this time $t_L(t_R)$. Without the perturbation, this time is given by $t_L(t_R) = -t_R - \tilde{\beta}/2$ where $\tilde{\beta}$ is defined in (5). To see this, note that since $r_* = 0$ on the boundary, an ingoing radial light ray from the right has $v = t_R$. Since $v = t + r_*$ is constant, and $t$ increases by $\tilde{\beta}/4$ at the singularity, we must have $r_* = -\tilde{\beta}/4$ at the singularity. An ingoing light ray from the left that meets it at the singularity has constant $u = v - 2r_* = t_R + \tilde{\beta}/2$. This is the time on the left boundary, but since we are requiring that that time increase to the future, we have $t_L = -u = -t_R - \tilde{\beta}/2$.

With the shock wave present, the only difference is that $v$ increases across the shock due to the jump in the Kruskal coordinate $V$. As a result, the left time becomes

$$t_L(t_R) = -t_R - \frac{\tilde{\beta}}{2} - \frac{\beta}{2\pi}\log\left(1 + \alpha e^{-\frac{2\pi}{\beta}t_R}\right). \tag{13}$$

We again see that when $t_R$ is large positive, the effect of the shock wave is negligible. As we decrease $t_R$, the last term in (13) becomes more and more important. Eventually, $t_L(t_R)$ approaches a constant $-\frac{\tilde{\beta}}{2} - \frac{\beta}{2\pi}\log\alpha$. This is another manifestation of the fact that the singularity bends down compared with the case without shock wave, and the amount of bending down depends on $t_R$.

Our goal in this paper is to extract $t_L(t_R)$ (13) from a particular form of the left-right correlator.

# 3 Outline of calculation and main results

Since the calculation in Section 4 is rather long, in this section we give an outline, describing the main steps and results.

We will do concrete calculations for a Schwarzschild-AdS black hole with a shock wave on the horizon in dimension $D = 5$, though we expect our result to hold for more general dimensions $D > 3$. We consider a scalar field with mass $m$ in this background. Since the shock wave breaks the boost symmetry, a left-right correlator will depend on two times, one from each boundary. The dual field theory description starts with

$$G(t_L, t_R) = \langle \text{TFD}| \psi_L(t_w) \mathcal{O}_L(t_L) \mathcal{O}_R(t_R) \psi_L(t_w) |\text{TFD}\rangle \,, \tag{14}$$

where $\psi_L(t_w)$ is an operator with conformal dimension $\Delta_\psi \gg \Delta_\mathcal{O}$ which creates a shock wave at left time $t_w$ with energy $E$. We consider the left-right correlator of the operator $\mathcal{O}$ dual to $\phi$ which has conformal dimension

$$\Delta_\mathcal{O} = \frac{D-1}{2} + \nu, \qquad \nu = \sqrt{\frac{(D-1)^2}{4} + m^2}. \tag{15}$$

We then take the limit $t_w \to -\infty$, $E \to 0$ keeping $\alpha \sim \frac{E}{M} e^{-\frac{2\pi}{\beta} t_w}$ fixed. Next, we Fourier transform to get $G(\omega_L, \omega_R)$.[7]

To calculate $G(\omega_L, \omega_R)$, we first solve the equations of motion for the modes on the shock wave background. We start with a Hartle-Hawking state on right, and propagate it across the shock wave. A complete set of modes is given in (34). Using this mode expansion, we obtain the general form of $G(\omega_L, \omega_R)$ (38). We do a detailed analysis of the analytic properties of $G(\omega_L, \omega_R)$ to justify our later analytic continuation to imaginary frequency.

Next we evaluate various quantities appearing in the two-point function (38). These quantities are obtained through solutions of the wave equation, which can be solved in a WKB approximation in the large mass limit. The solutions have the form of integrals of various bulk quantities (62). When we take the frequencies imaginary, the integrals in (62) can be related to properties of spacelike geodesics connecting left and right boundaries. The bouncing geodesic as discussed in [9] (see review in Section 2) corresponds to large imaginary frequencies. We have been suppressing the angular mode labels in $G$, but it will suffice to consider the spherically symmetric mode since this is related to radial geodesics like the bouncing geodesic we wish to recover.

As our goal is to quantify how much the singularity bends down as a function of $t_R$, we do a Fourier transform in $\omega_R$ back to $t_R$, while keeping $\omega_L$ negative imaginary and large.[8]

$$G(\omega_L, t_R) \equiv \int_{-\infty}^{+\infty} \frac{d\omega_R}{2\pi} e^{-i\omega_R t_R} G(\omega_L, \omega_R). \tag{16}$$

This is a novel form of the correlator which we will see expands upon [9] and [10]. One nice feature of this mixed frequency-time correlator is that, when doing the Fourier transform in $\omega_R$ using the method of steepest descent, we actually pick up the saddle corresponding to the bouncing geodesic in real spacetime. This might seem surprising at first, since the position space correlator is not dominated by this bouncing geodesic. The reason for this difference

---

[7]Our convention for the momentum space correlator is that

$$G(\omega_L, \omega_R) = \int_{-\infty}^{+\infty} dt_L dt_R e^{i\omega_L t_L + i\omega_R t_R} G(t_L, t_R).$$

[8]How large is large enough will depend on $t_R$. See Section 4.4 for details.

is simply the hybrid form of the correlator, with large imaginary $\omega_L$. The resulting two-point function has the following behavior (108):

$$G(\omega_L = -i\tilde{E}_L, t_R) \sim e^{\tilde{E}_L t_L(t_R)} = e^{-\tilde{E}_L \left[t_R + \frac{\tilde{\beta}}{2} + \frac{\beta}{2\pi} \log\left(1 + \alpha e^{-\frac{2\pi}{\beta} t_R}\right)\right]}. \tag{17}$$

So the coefficient of the exponential behavior of the hybrid correlator at large imaginary $\omega_L$ is precisely $t_L(t_R)$ (13).

# 4 Calculations

In this section, we present the calculations that lead to our main result (17). In Section 4.1, we first derive a general expression for the two-point function via a mode expansion and study some of its properties. Since the general expression is not known exactly,[9] we derive a large mass approximation in Section 4.2. In Section 4.3, we discuss the analytic continuation of the large mass expression to complex frequencies, which reveals a concrete relation between the two-point function and geodesics in the shock wave spacetime.[10] Finally in Section 4.4, we Fourier transform the large mass expression in one of the frequencies to time and obtain the result (17).

## 4.1 Two-point function of a massive scalar field in a shock wave spacetime: General expression

We will calculate the two-point function by a mode expansion of $\phi$. We first focus on the field equation in the absence of the shock wave. Consider a solution of the form $\phi = e^{-i\omega t} Y_I(e) r^{-\frac{D-2}{2}} \psi_{\omega,l}(r)$, where $e$ denotes the angular coordinates and $Y_I$ are the spherical harmonics on $S^{D-2}$, with $I$ denoting the full set of indices including the angular momentum $l$, so that $\nabla^2_{S^{D-2}} Y_I = -l(l + D - 3)Y_I$. The equation of motion for $\phi$ in terms of the tortoise coordinate $r_*$ (4) takes the form of the following Schrodinger equation

$$(-\partial^2_{r_*} + U_l(r_*) - \omega^2)\psi_{\omega,l} = 0, \tag{18}$$

where

$$U_l(r_*) = f(r)\left[\frac{(2l + D - 3)^2 - 1}{4r^2} + \nu^2 - \frac{1}{4} + \frac{(D-2)^2\mu}{4r^{D-1}}\right]. \tag{19}$$

The potential $U_l$ behaves near the boundary $r_* \to 0$ as

$$U_l \approx \frac{\nu^2 - \frac{1}{4}}{r_*^2}, \tag{20}$$

and near the horizon $r_* \to -\infty$ as

$$U_l \propto e^{\frac{4\pi r_*}{\beta}}. \tag{21}$$

We can then normalize $\psi_{\omega l}$ at the horizon by requiring

$$\psi_{\omega,l} \approx e^{i\omega r_* - i\delta_{\omega,l}} + e^{-i\omega r_* + i\delta_{\omega,l}}, \tag{22}$$

---

[9]See however [24] for a recent derivation of the exact retarded propagator in Schwarzschild-AdS without perturbations.

[10]The analytic continuation explained in Section 4.3.1 contains some technical details which are important for the relation to geodesics described in Section 4.3.2, but readers not interested in the details may skip directly to Section 4.3.2.

where $\delta_{\omega,l}$ is real for $\omega > 0$. The form of $\psi_{\omega l}$ at the boundary is

$$\psi_{\omega,l} \approx C(\omega,l)(-r_*)^{\nu+1/2}\,, \tag{23}$$

where $C(\omega,l)$ is fixed by the normalization. We also define modes that are supported on the left and right regions

$$\phi_{\omega,l}^{R,L}(t,r,e) = e^{-i\omega t}Y_I(e)r^{-\frac{D-2}{2}}\psi_{\omega,l}(r)\,. \tag{24}$$

In the absence of the shock wave, the Hartle-Hawking state is defined by taking modes of the following form to be the positive frequency modes [25]:[11]

$$H_{\omega,l}^{(1)} = \sqrt{\frac{1}{1-e^{-\beta\omega}}}\phi_{\omega,l}^R + \sqrt{\frac{e^{-\beta\omega}}{1-e^{-\beta\omega}}}\phi_{\omega,l}^L\,,$$
$$H_{\omega,l}^{(2)} = \sqrt{\frac{1}{e^{\beta\omega}-1}}\phi_{\omega,l}^{R*} + \sqrt{\frac{e^{\beta\omega}}{e^{\beta\omega}-1}}\phi_{\omega,l}^{L*}\,. \tag{25}$$

With the shock wave, we define our state by requiring that the modes $H_{\omega,l}^{(1,2)}$ restricted to the right exterior to continue to be our positive frequency modes, but their continuation into the left is determined by the field equation in the presence of the shock wave.

When the effect of the shock wave is included, the metric becomes [26][12]

$$ds^2 = \frac{f(r(U,\tilde{V}))}{\kappa^2 U\tilde{V}}dUd\tilde{V} - \alpha\frac{f(r(U,\tilde{V}))}{\kappa^2 U\tilde{V}}\delta(U)dU^2 + r(U,\tilde{V})^2 d\Omega^2\,. \tag{26}$$

The field equation for the scalar field can be written as

$$(\nabla_0^2 - m^2)\phi = -\frac{16\pi^2\alpha U\tilde{V}}{\beta^2 f}\delta(U)\partial_{\tilde{V}}^2\phi\,, \tag{27}$$

where $\nabla_0^2$ is the Laplacian operator of the metric without the shock wave and the term on the right includes all the effects of the shock wave. Near the horizon, this becomes

$$\partial_U\partial_{\tilde{V}}\phi = -\alpha\delta(U)\partial_{\tilde{V}}^2\phi\,. \tag{28}$$

Solving this, we get

$$\phi(U=0^+,\tilde{V}) = \phi(U=0^-,\tilde{V}-\alpha)\,. \tag{29}$$

Note that the two sides of equation (29) are to be compared at the same value of $\tilde{V}$, so in terms of the continuous $V$ coordinate, both sides are evaluated at $V = \tilde{V} - \alpha$, i.e. $\phi$ is continuous across the shock. This must be the case since the near-horizon metric is simply $ds^2 \approx -4dUdV + r_0^2 d\Omega_{D-2}^2$ in continuous $U, V$ coordinates.

Our positive frequency modes $\tilde{H}_{\omega_R,l}^{(1,2)}$ restricted to the right are exactly the same as $H_{\omega_R,l}^{(1,2)}$ and are proportional to $\phi_{\omega,l}^R$ and $\phi_{\omega,l}^{R*}$. Near the right future horizon before the shock wave, we have

$$H_{\omega,l}^{(1)} \approx \sqrt{\frac{1}{1-e^{-\beta\omega}}}Y_I(e)r_0^{-\frac{D-2}{2}}\left(\tilde{V}^{-i\frac{\beta}{2\pi}\omega}e^{i\delta_\omega} + (-U)^{i\frac{\beta}{2\pi}\omega}e^{-i\delta_\omega}\right)\,,$$
$$H_{\omega,l}^{(2)} \approx \sqrt{\frac{1}{e^{\beta\omega}-1}}Y_I(e)r_0^{-\frac{D-2}{2}}\left(\tilde{V}^{i\frac{\beta}{2\pi}\omega}e^{-i\delta_\omega} + (-U)^{-i\frac{\beta}{2\pi}\omega}e^{i\delta_\omega}\right)\,. \tag{30}$$

---

[11]Note that the choice of relative coefficients between $\phi^R$ and $\phi^L$ amounts to take $t \to t - i\frac{\beta}{2}$ on the left. In what sense they are positive frequency modes will be explained shortly.

[12]Here we are using the discontinuous coordinate $\tilde{V} = V + \alpha\Theta(U)$ in which the metric has a delta function.

These are positive frequency modes in the sense that as complex functions of $U$ and $V$, they are analytic in the lower half plane, i.e., we put the branch cut on the upper half plane, or $-1 = e^{-i\pi}$. As a consequence, when we go to the left past horizon where $\tilde{V} < 0$ and $U > 0$, (30) becomes

$$
\begin{aligned}
H^{(1)}_{\omega,l} &\approx \sqrt{\frac{e^{-\beta\omega}}{1-e^{-\beta\omega}}} Y_I(e) r_0^{-\frac{D-2}{2}} \left( (-\tilde{V})^{-i\frac{\beta}{2\pi}\omega} e^{i\delta_\omega} + U^{i\frac{\beta}{2\pi}\omega} e^{-i\delta_\omega} \right), \\
H^{(2)}_{\omega,l} &\approx \sqrt{\frac{e^{\beta\omega}}{e^{\beta\omega}-1}} Y_I(e) r_0^{-\frac{D-2}{2}} \left( (-\tilde{V})^{i\frac{\beta}{2\pi}\omega} e^{-i\delta_\omega} + U^{-i\frac{\beta}{2\pi}\omega} e^{i\delta_\omega} \right).
\end{aligned}
\tag{31}
$$

Note that (31) is also consistent with the definition (25) when restricting to the left exterior.

The $U$-dependent terms oscillate as $U \to 0$ and vanish when smeared in frequency. We will focus on the $V$-dependent terms. We then use the matching condition to take $(-\tilde{V})^{\pm i\frac{\beta}{2\pi}\omega}$ to the left across the shock wave. From (29), on the left immediately across the shock wave, $(-\tilde{V})^{-i\frac{\beta}{2\pi}\omega}$ becomes

$$
(-\tilde{V} + \alpha)^{-i\frac{\beta}{2\pi}\omega_R} = \int \frac{d\omega_L}{2\pi} T_{\omega_L,\omega_R} (-\tilde{V})^{-i\frac{\beta}{2\pi}\omega_L},
\tag{32}
$$

where

$$
T_{\omega_L,\omega_R} = \frac{\beta}{2\pi} \frac{\alpha^{i\frac{\beta}{2\pi}(\omega_L-\omega_R)}}{\Gamma\left(i\frac{\beta}{2\pi}\omega_R\right)} \Gamma\left(-i\frac{\beta}{2\pi}(\omega_L-\omega_R)\right) \Gamma\left(i\frac{\beta}{2\pi}\omega_L\right).
\tag{33}
$$

The form of $T_{\omega_L,\omega_R}$ is obtained by noting that (32) is a Fourier transform when written in terms of Eddington-Finkelstein coordinates. However, making eq. (32) precise requires a contour prescription that we explain in detail in Appendix A.1. The positive frequency modes for our state are therefore

$$
\begin{aligned}
\tilde{H}^{(1)}_{\omega_R,l} &= \sqrt{\frac{1}{1-e^{-\beta\omega_R}}} \phi^R_{\omega_R,l} + \sqrt{\frac{e^{-\beta\omega_R}}{1-e^{-\beta\omega_R}}} \int \frac{d\omega_L}{2\pi} e^{i(\delta_{\omega_R}-\delta_{\omega_L})} T_{\omega_L,\omega_R} \phi^L_{\omega_L,l}, \\
\tilde{H}^{(2)}_{\omega_R,l} &= \sqrt{\frac{1}{e^{\beta\omega_R}-1}} \phi^{R*}_{\omega_R,p} + \sqrt{\frac{e^{\beta\omega_R}}{e^{\beta\omega_R}-1}} \int \frac{d\omega_L}{2\pi} e^{-i(\delta_{\omega_R}-\delta_{\omega_L})} T^*_{\omega_L,\omega_R} \phi^{L*}_{\omega_L,l}.
\end{aligned}
\tag{34}
$$

The state we are interested in is the vacuum state with respect to these modes.

We can expand the field $\phi$ in terms of these modes as

$$
\phi = \sum_l \int_0^\infty \frac{d\omega}{2\pi} \left( \tilde{H}^{(1)}_{\omega,l} b^{(1)}_{\omega,l} + \tilde{H}^{(2)}_{\omega,l} b^{(2)}_{\omega,l} \right) + c.c.,
\tag{35}
$$

where $b^{(1)}_{\omega,l}$ and $b^{(2)}_{\omega,l}$ are the annihilation operators of the corresponding modes, normalized as

$$
\left[ b^{(i)}_{\omega,l}, b^{(j)\dagger}_{\omega',l'} \right] = \frac{2\pi}{2\omega} \delta(\omega-\omega') \delta_{l,l'} \delta_{i,j}.
\tag{36}
$$

We then obtain the bulk two-point function as

$$
\begin{aligned}
\mathcal{G}(\omega_L,\omega_R,l;r_L,r_R) = \frac{1}{4\pi} \frac{\beta^2}{(2\pi)^2} \alpha^{-i\frac{\beta}{2\pi}(\omega_L-\omega_R)} \Gamma\left(\frac{\beta}{2\pi}i\omega_R\right) \Gamma\left(-i\frac{\beta}{2\pi}\omega_L\right) \Gamma\left(i\frac{\beta}{2\pi}(\omega_L-\omega_R)\right) \\
\times e^{i(-\delta_{\omega_R,l}+\delta_{\omega_L,l})} \psi_{\omega_L,l}(r_L) \psi_{\omega_R,l}(r_R)(r_L r_R)^{-\frac{D-2}{2}}.
\end{aligned}
\tag{37}
$$

The boundary two-point function can then be obtained from the extrapolation dictionary.

$$
\begin{aligned}
G(\omega_L, \omega_R, l) &= \lim_{r_L, r_R \to \infty} (2 v r_R^\Delta)(2 v r_L^\Delta) \mathcal{G}(\omega_L, \omega_R, l; r_L, r_R) \\
&= \frac{v^2 \beta^2}{(2\pi)^2 \pi} \alpha^{-i\frac{\beta}{2\pi}(\omega_L - \omega_R)} \Gamma\left(\frac{\beta}{2\pi} i \omega_R\right) \Gamma\left(-i\frac{\beta}{2\pi} \omega_L\right) \Gamma\left(i\frac{\beta}{2\pi}(\omega_L - \omega_R)\right) \\
&\quad \times e^{-i\delta_{\omega_R, l}} C(\omega_R, l) e^{i\delta_{\omega_L, l}} C(\omega_L, l).
\end{aligned}
\tag{38}
$$

Notice that the form of (38) only depends on the Kruskal coordinate shift at the horizon. The details of the metric are encoded in the functions $e^{\pm i\delta_{\omega, l}} C(\omega, l)$.

### 4.1.1 Analytic properties of the momentum two-point function

We briefly summarize the analytic properties of $G(\omega_L, \omega_R)$ in the complex $\omega_L$ and $\omega_R$ plane. This will be crucial in determining the analytic continuation of the large frequency expressions derived in Section 4.2. It is easy to see that the gamma functions give rise to lines of equally spaced first order poles in both planes. In the $\omega_R$ plane, there is one line along the positive imaginary axis from the origin, and another extending from $\omega_L$ in the negative imaginary direction

$$
\omega_R = \begin{cases} i\frac{2\pi n}{\beta}, & n = 0, 1, \dots, \\ \omega_L - i\frac{2\pi m}{\beta}, & m = 0, 1, \dots \end{cases}
\tag{39}
$$

The case in the $\omega_L$ plane is similar, with one line along the negative imaginary axis from the origin, and another extending from $\omega_R$ in the positive imaginary direction

$$
\omega_L = \begin{cases} -i\frac{2\pi n}{\beta}, & n = 0, 1, \dots, \\ \omega_R + i\frac{2\pi m}{\beta}, & m = 0, 1, \dots \end{cases}
\tag{40}
$$

Deducing the analyticity properties of $e^{-i\delta_{\omega_R l}} C(\omega_R, l)$ and $e^{i\delta_{\omega_L l}} C(\omega_L, l)$ is more involved, and we simply state the results and leave details of the argument to Appendix A.2. We will also often consider only the case of $l = 0$ and the labels for $l$ will be dropped whenever we restrict to this case. $e^{-i\delta_{\omega_R}} C(\omega_R)$ ($l = 0$) as a function of $\omega_R$ has the following properties.

1. It has a reflection symmetry about the imaginary axis

$$
e^{-i\delta_{-\omega_R^*}} C(-\omega_R^*) = (e^{-i\delta_{\omega_R}} C(\omega_R))^*.
\tag{41}
$$

2. The only singularities it has are poles in the upper half plane and correspond to the quasinormal frequencies reflected by the real axis. These poles come in pairs reflected about the imaginary axis. The location of these poles are well-studied and are lines of poles lying off the imaginary axis [27, 28].

3. It has a line of zeroes along the positive imaginary axis $\omega_R = i\frac{2\pi n}{\beta}$, $n = 0, 1, \dots$ . These cancel all the poles along the positive imaginary axis from the gamma function (39) when combined in $G(\omega_L, \omega_R)$.

Properties of $e^{i\delta_{\omega_L}} C(\omega_L)$ are obtained from those of $e^{-i\delta_{\omega_R}} C(\omega_R)$ by taking $\omega_R \to \omega_L$ and reflecting about the real axis. In particular, it has a line of zeroes that cancel with the poles along the negative imaginary axis in (40) and has poles precisely at the quasinormal frequencies.

In summary, $G(\omega_L, \omega_R)$ in the $\omega_R$ plane has lines of poles at the reflection of quasinormal frequencies in the upper half plane, and a line of poles extending from $\omega_L$ in the negative imaginary direction (see Figure 3); in the $\omega_L$ plane, it has lines of poles at the quasinormal frequencies in the lower half plane, and a line of poles extending from $\omega_R$ in the positive imaginary direction.

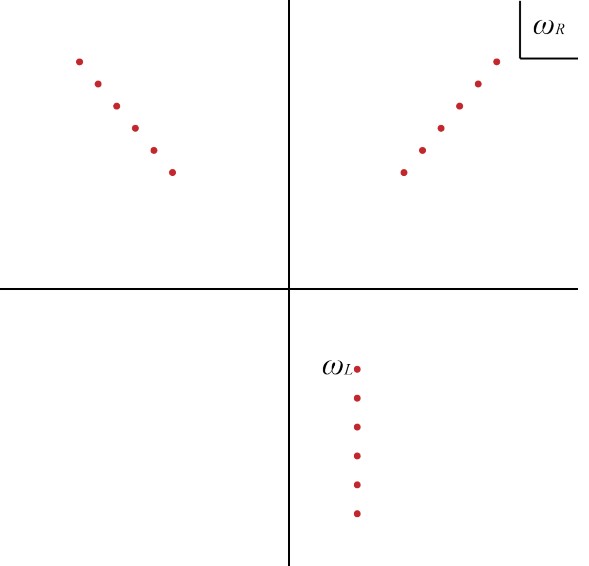

Figure 3: The analytic structure of $G(\omega_L, \omega_R)$ in the $\omega_R$ plane for a fixed complex $\omega_L$. The poles in the upper half plane are the reflection of the quasinormal modes and those in the lower half plane start at the fixed complex $\omega_L$.

### 4.1.2 A "thermal" product formula in a shock wave background

Recently, it was pointed out in [21] that holographic thermal two-point functions take the form of a product over quasinormal modes. The proof relies mostly on the fact that the boundary two-point Wightman function $G_{\text{thermal}}(\omega)$ for a static spherically symmetric black hole is meromorphic, since it has only isolated simple poles, and that $1/G_{\text{thermal}}(\omega)$ is entire, which follows from the fact that $G_{\text{thermal}}(\omega)$ itself has no zeros - a feature that is connected to the presence of an horizon. Given these properties, together with a few other technical details, one can use the Hadamard factorization theorem[13] to show that [21]

$$G_{\text{thermal}}(\omega) = \frac{G_{\text{thermal}}(0)}{\prod_{n=1}^{\infty}\left(1 - \frac{\omega^2}{\omega_n^2}\right)\left(1 - \frac{\omega^2}{(\omega_n^*)^2}\right)}, \tag{42}$$

where, for concreteness, we take $\omega_n$ to correspond to the quasinormal modes on the right side of the lower half-plane. We emphasize that $\omega_n$ does not denote the full set of quasinormal modes. Due to the properties of $G_{\text{thermal}}$, its poles come in families $(\omega_n, -\omega_n, \omega_n^*, -\omega_n^*)$. We will be using $\omega_n$ and $-\omega_n^*$ to denote the poles in the lower half-plane of $G_{\text{thermal}}(\omega)$ which correspond precisely to the full set of quasinormal modes (these are in fact the reflection with respect to the real axis of the poles located in the upper half-plane of Figure 3). As pointed out in [21], the expression (42) does not account for the presence of purely imaginary quasinormal modes, even though they can be readily included. Since we are mainly focusing on Schwarzschild-AdS while restricting to $l = 0$ modes, we do not have purely imaginary quasinormal modes and thus we will not account for them in what follows. See Appendix B for details on the more general case.

Below we present an analogous decomposition for the two-point function in the shock wave background we are studying. First, notice that the two-point function we computed

---

[13]See Appendix B for details on the Hadamard factorization theorem.

in (38) can be written as[14]

$$G(\omega_L, \omega_R) = \frac{\nu^2 \beta^2}{(2\pi)^2 \pi} \Gamma\left(\frac{\beta}{2\pi} i\Delta\omega\right) \alpha^{-i\frac{\beta}{2\pi}\Delta\omega} G_L(\omega_L) G_R(\omega_R), \tag{43}$$

where we defined

$$G_L(\omega_L) = \Gamma\left(-\frac{\beta}{2\pi} i\omega_L\right) e^{i\delta_{\omega_L}} C(\omega_L), \tag{44}$$

$$G_R(\omega_R) = \Gamma\left(\frac{\beta}{2\pi} i\omega_R\right) e^{-i\delta_{\omega_R}} C(\omega_R), \tag{45}$$

with $\Delta\omega = \omega_L - \omega_R$. The above functions are meromorphic and have no zeros, i.e. $1/G_{L/R}(\omega)$ are entire functions. These properties follow from the analytic properties described in 4.1.1 and correspond exactly to the main features that allowed [21] to use the Hadamard factorization theorem to write (42). As a result, both $G_L$ and $G_R$ admit appropriate Hadamard factorizations whose details are spelled out in Appendix B. Applying the Hadamard factorization to $G_R(\omega_R)$, $G_L(\omega_L)$ and $\Gamma\left(\frac{\beta}{2\pi} i\Delta\omega\right)$ in (43), we find that we can write[15]

$$G(\omega_L, \omega_R) = \frac{G_{\text{thermal}}(0) e^{ig_\alpha(\Delta\omega)}}{\Delta\omega \prod_{n=1}^{\infty}\left(1 - \frac{\Delta\omega}{i\Omega_n}\right)\left(1 - \frac{\omega_L}{\omega_n}\right)\left(1 + \frac{\omega_L}{\omega_n^*}\right)\left(1 + \frac{\omega_R}{\omega_n}\right)\left(1 - \frac{\omega_R}{\omega_n^*}\right)}, \tag{46}$$

with

$$g_\alpha(\Delta\omega) = \frac{\beta}{2\pi}\Delta\omega(c - \gamma - \log\alpha) - \Delta\omega \sum_{n=1}^{\infty}\left(2\,\text{Im}\left\{\frac{1}{\omega_n}\right\} - \frac{1}{\Omega_n}\right) - \frac{\pi}{2}, \tag{47}$$

where $\Omega_n = 2\pi n/\beta$ are the Matsubara frequencies, $\gamma$ is the Euler's constant, $\text{Im}\{1/\omega_n\}$ denotes the imaginary part of $1/\omega_n$ and

$$c = i\frac{2\pi}{\beta}\frac{G_R'(0)}{G_R(0)} = -i\frac{2\pi}{\beta}\frac{G_L'(0)}{G_L(0)}. \tag{48}$$

Using (46), we see that all the dependence of this two-point function on the geometry can mostly be reduced to its quasinormal modes, the Matsubara frequencies and the parameter $\alpha$ which characterizes the shock wave. There are however two undetermined constants: an overall rescaling $G_{\text{thermal}}(0)$ which was already present in (42) and a real constant $c$ which only changes $g_\alpha$. Furthermore, the form (46) makes the analytic structure of $G(\omega_L, \omega_R)$ fully transparent.

Finally, we note that (46) implies that the thermal two-point function is captured by the residue of our two-point function at $\Delta\omega = 0$. Namely,

$$\text{Res}_{\Delta\omega=0}\, G(\omega_L, \omega_R) = -iG_{\text{thermal}}(\omega), \tag{49}$$

where we set $\omega_R = \omega_L = \omega$. This is a natural consequence of the fact that our expression should reduce to the thermal two-point function in the limit $\alpha \to 0$ with an appropriate choice of contour. When $\alpha \to 0$, we have $g_\alpha(\Delta\omega) \sim -\Delta\omega \log\alpha$ because, by the Hadamard factorization theorem, the infinite sum in $g_\alpha$ together with the infinite product in the denominator

---

[14]The notation $G_R$ is frequently used to denote the retarded propagator. Here we use the label $R$ just to denote the fact that it is a function which only depends on $\omega_R$.

[15]We note that, while it is possible to prove that (46) must hold whenever (42) holds in the underlying geometry without a shock, the form (47) relies on $1/G_{L/R}(\omega)$ being entire functions of finite order 1. Our WKB results (62) strongly suggest this is true generically and one can refer to results in [29] or our own (100) to readily check it is true in Schwarzschild-AdS in $D = 5$. For more details on the technical aspects involved in deriving (46) and (47) see Appendix B.

should converge and the remaining terms in $g_\alpha$ are finite. Consider doing the inverse Fourier transform of (46) in $\omega_L$, while keeping $\omega_R \in \mathbb{R}$. There is a pole on the real axis at $\omega_L = \omega_R$ and we choose to Fourier transform by picking a contour $\gamma$ that goes below it.[16] Due to the behavior of $g_\alpha$, for $\alpha \ll 1$, the behavior of our integrand on the upper half-plane is controlled by $G(\omega_L, \omega_R)e^{-i\omega_L t_L} \sim e^{-i\omega_L(\frac{\beta}{2\pi}\log\alpha + t_L + C(\omega_n, \Omega_n))}$, where $C(\omega_n, \Omega_n)$ is some constant, which depends on the Matsubara frequencies and the quasinormal modes. This means that, given a fixed $t_L$, we can always make $\alpha$ small enough such that the integrand goes to zero on a semi-circle in the upper half-plane which we can use to close our contour. Thus, the inverse Fourier transform reduces to a sum over residues at $\Delta\omega = i\tilde{\omega}_n$ with $n \in \mathbb{N}_0$, since these are the poles that are present in the upper half plane of $\omega_L$. In the limit $\alpha \to 0$, only the residue at $\Delta\omega = 0$ contributes due to the behavior $g_\alpha(\Delta\omega) \sim -\Delta\omega\log\alpha$. It thus follows that

$$\lim_{\alpha\to 0}\int_\gamma \frac{d\omega_L}{2\pi}G(\omega_L, \omega_R)e^{-i\omega_L t_L} = G_{\text{thermal}}(\omega_R)e^{-i\omega_R t_L}. \tag{50}$$

The factor of $e^{-i\omega_R t_L}$ is simply a consequence of the fact that we are coming from the finite $\alpha$ result where the boost symmetry was broken. Doing the inverse Fourier transform in $\omega_R$ restores the expected result by combining this extra Fourier mode with $e^{-i\omega_R t_R}$, yielding

$$G_{\text{thermal}}(t) = \int \frac{d\omega}{2\pi}G_{\text{thermal}}(\omega)e^{-i\omega t}, \tag{51}$$

where we identified $t = t_L + t_R$.

We will not make further use of the decomposition (46) in our work. However, given the similarities to (42), it would be interesting to understand if and how some of the properties derived in [21] extend to our case. More ambitiously, given the relationship between this two point function and the thermal four-point function (14), it would be interesting to concretely determine the regimes of validity of this kind of decomposition for holographic thermal correlators more generally [30, 31].

## 4.2 Large mass limit

In this section, we will study $G(\omega_L, \omega_R, l)$ in a large mass limit. It is useful to define

$$\omega = \nu u, \qquad l + \frac{D-3}{2} = \nu k, \tag{52}$$

and consider $G(\omega_L, \omega_R, l)$ as a function of $u$ and $k$ in the limit of large $\nu$. In this limit, the lines of poles in the $\omega$ planes mentioned in Section 4.1.1 become branch cuts in the $u$ planes and play an important role in the analytic continuation.

### 4.2.1 Matching function

It will be convenient to aligned $G(\omega_L, \omega_R, l)$ into two groups of factors. We will first consider the following factors, which come from matching the solution to the wave equation across the shock wave.

$$\frac{\nu^2\beta^2}{(2\pi)^2\pi}\alpha^{-i\frac{\beta}{2\pi}(\omega_L-\omega_R)}\Gamma\left(i\frac{\beta}{2\pi}\omega_R\right)\Gamma\left(-i\frac{\beta}{2\pi}\omega_L\right)\Gamma\left(i\frac{\beta}{2\pi}(\omega_L-\omega_R)\right). \tag{53}$$

---

[16]This choice is fundamentally tied to the definition of $T_{\omega_L,\omega_R}$ explained in A.1, which relies on an $i\epsilon$ prescription that propagates to the definition of the frequency space two-point function. In fact, this calculation that we are presenting resembles the one in Appendix A.1.

We apply Stirling's formula for large $\nu$ to get

$$\frac{2\nu^{1/2}\beta^{1/2}}{\sqrt{iu_R u_L(u_L - u_R)}}\exp\left[\nu\left(\frac{i\beta u_R}{2\pi}\log\left(\frac{\alpha u_R}{u_L - u_R}\right) + \frac{i\beta u_L}{2\pi}\log\left(\frac{u_L - u_R}{\alpha u_L}\right) - \frac{\beta}{2}u_L\right)\right]. \tag{54}$$

This expression has branch points at $u_R = 0$, $u_L = 0$ and $u_R = u_L$. The branch cuts of the logarithms are chosen to coincide with the location of the line of poles in (53), i.e. in the $u_R$ plane, there is a cut along the positive imaginary axis from $u_R = 0$ and one extending from $u_L$ in the negative imaginary direction; in the $u_L$ plane, there is a cut along the negative imaginary axis from $u_L = 0$ and one extending from $u_R$ in the positive imaginary direction. Given the results in 4.1.1, we expect the branch cuts extending from the origin in both planes to be cancelled by contributions arising from the remaining factors. We will see that this is indeed the case and the final results have the expected analytic structure.

### 4.2.2 Modes from WKB approximation

The remaining factors are

$$e^{-i\delta_{\omega_R l}}C(\omega_R, l)e^{i\delta_{\omega_L l}}C(\omega_L, l) = e^{-i\delta_{\omega_R l}}e^{i\delta_{\omega_L l}}\lim_{r_L, r_R \to \infty}(r_L r_R)^{\Delta - \frac{D-2}{2}}\psi_{\omega_L l}(r_L)\psi_{\omega_R l}(r_R). \tag{55}$$

All of these quantities are related to the Schrodinger problem (18), and we can use the WKB approximation to obtain a large mass expression. Note that although the position Wightman function involves $G(\omega_L, \omega_R, l)$ for all real $\omega_L, \omega_R$, it suffices to explicitly calculate these quantities for $\omega_L, \omega_R > 0$ due to the reflection property mentioned in 4.1.1. Writing $\psi_{\omega, l} = e^{\nu S}$ in (18) gives

$$-(\partial_{r_*}S)^2 - \frac{1}{\nu}\partial_{r_*}^2 S + V(r_*) + \frac{1}{\nu^2}Q(r_*) = u^2, \tag{56}$$

where

$$V(r_*) = f(r)\left(1 + \frac{k^2}{r^2}\right), \qquad Q(r_*) = f(r)\left(-\frac{1}{4r^2} - \frac{1}{4} + \frac{(D-2)^2\mu}{4r^{D-1}}\right). \tag{57}$$

We solve the expansion up to $\mathcal{O}(1/\nu)$ using the usual WKB methods. Imposing the normalization condition of (22), we obtain

$$\psi_{\omega, l}(r) = \begin{cases} \frac{\sqrt{u}}{(V(r) - u^2)^{1/4}}\exp\left(-\nu\int_{r_c}^{r}\frac{dr}{f(r)}\sqrt{V(r) - u^2}\right), & r > r_c, \\ \frac{2\sqrt{u}}{(u^2 - V(r))^{1/4}}\cos\left(\nu\int_{r}^{r_c}\frac{dr}{f(r)}\sqrt{u^2 - V(r)} - \frac{\pi}{4}\right), & r < r_c, \end{cases} \tag{58}$$

where $r_c(u)$ is the turning point satisfying $u^2 = V(r_c)$. Using the expression in the classically forbidden region, we have, for large $\nu$

$$\lim_{r \to \infty}(r)^{\nu + \frac{1}{2}}\psi_{\omega, l}(r) \approx \sqrt{u}\lim_{r \to \infty}\exp\left[\nu\left(\log r - \int_{r_c}^{r}\frac{dr}{f(r)}\sqrt{V(r) - u^2}\right)\right]. \tag{59}$$

The remaining factors $e^{\pm i\delta_{\omega, l}}$ are related to the WKB solution in the classically allowed region since they are defined at the horizon via (22). Comparing the two expressions gives

$$\begin{aligned} e^{i\delta_{\omega, l}} &= \lim_{r \to r_0}\exp\left[i\nu\int_{r}^{r_c}\frac{dr}{f(r)}\sqrt{u^2 - V(r)} - i\frac{\pi}{4} + i\omega r_*(r)\right] \\ &= e^{-i\frac{\pi}{4}}\lim_{r \to r_0}\exp\left[\nu\left(i\int_{r}^{r_c}\frac{dr}{f(r)}\sqrt{u^2 - V(r)} - iu\int_{r}^{\infty}\frac{dr}{f(r)}\right)\right]. \end{aligned} \tag{60}$$

Evaluating (55) with the WKB expressions above and combining with (54), one obtains at $l = 0$

$$\lim_{\nu \to +\infty} G(\nu u_L, \nu u_R, 0) \approx \frac{2\nu^{1/2}\beta^{1/2}}{\sqrt{i(u_L - u_R)}} e^{\nu Z(u_L, u_R)}, \tag{61}$$

with

$$Z(u_L, u_R) = Z_\psi(u_R) + Z_\psi(u_L) - Z_\delta(u_R) + Z_\delta(u_L) + Z_{matching}(u_L, u_R),$$

$$Z_\psi(u_a) = \lim_{r \to \infty} \left( \log r - \int_{r_c(u_a)}^{r} \frac{dr'}{f(r')} \sqrt{f(r') - u_a^2} \right),$$

$$Z_\delta(u_a) = \lim_{r \to r_0} \left( i \int_{r}^{r_c(u_a)} \frac{dr'}{f(r')} \sqrt{u_a^2 - f(r')} - iu_a \int_{r}^{\infty} \frac{dr'}{f(r')} \right), \tag{62}$$

$$Z_{matching}(u_L, u_R) = \frac{i\beta u_R}{2\pi} \log \left( \frac{\alpha u_R}{u_L - u_R} \right) + \frac{i\beta u_L}{2\pi} \log \left( \frac{u_L - u_R}{\alpha u_L} \right) - \frac{\beta}{2} u_L,$$

where $a = L, R$, the $Z_\psi(u_a)$ terms are contributions from $C(\omega_L)$ and $C(\omega_R)$, the $Z_\delta(u_a)$ terms are contributions from $e^{i\delta_{\omega_L}}$ and $e^{-i\delta_{\omega_R}}$, and $Z_{matching}(u_L, u_R)$ comes from the matching function of Section 4.2.1.

One aspect of the WKB approximation of $G(\omega_L, \omega_R)$ that is important in this work is that it provides a clear connection between $G(\omega_L, \omega_R)$ and spacelike geodesics. Away from the shock wave, spacelike geodesics (with proper length $\lambda$) can be labelled by the conserved quantities related to the Killing vectors of time translation and rotation

$$E = f \frac{dt}{d\lambda}, \qquad q = r^2 \frac{d\theta}{d\lambda}. \tag{63}$$

The geodesics then satisfy

$$\frac{1}{f} \left( \frac{dr}{d\lambda} \right)^2 + \frac{q^2}{r^2} - \frac{1}{f} E^2 = 1, \tag{64}$$

which is exactly the field equation (56) at leading order in $\nu$ if we change variables to

$$\partial_{r_*} S \to -\frac{dr}{d\lambda}, \quad u \to -iE, \quad k \to -iq. \tag{65}$$

This suggests that $Z(u_L, u_R)$ at imaginary $u$ should have a simple connection to spacelike geodesics. However, $Z(u_L, u_R)$ is originally defined only for $u_L, u_R > 0$ and needs to be analytically continued to the entire complex plane. This involves some subtleties which will be the focus of Section 4.3.

## 4.3 Analytic continuation of large mass expression and relation to geodesics

In this section, we discuss how to analytically continue $Z(u_L, u_R)$ into the complex plane. Starting with only the integrals defining $Z(u_L, u_R)$ at $u_L, u_R > 0$, the analytic continuation is not unique. One needs additional input from the analytic structure of $G(\omega_L, \omega_R)$. In particular, we will pick the branch cuts of $Z(u_L, u_R)$ to match exactly with the lines of poles of $G(\omega_L, \omega_R)$ that were discussed in Section 4.1.1. Not only does this uniquely determine the analytic structure of $Z(u_L, u_R)$, it also guarantees that our large mass expression is a good approximation of $G(\omega_L, \omega_R)$ in the entire complex plane and not just for $u_L, u_R > 0$ where the derivation was done. The analytic continuation of $Z(u_L, u_R)$ allows $u_a$ to take values in the imaginary axis, thus making the connection to geodesics manifest.

### 4.3.1 Analytic continuation of large mass expression

**Analytic continuation of $r_c(u)$**

The analytic continuation of $Z$ involves two parts: (1) analytically continuing the turning point $r_c(u)$, which comes up in the limits of integration and (2) specifying contours around the pole of the integrand at $r = r_0$. We start with the analytic continuation of $r_c(u)$, which was studied in [10, 27, 29], and we simply summarize their results. At $u > 0$, $r_c(u)$ is the unique positive solution to $u^2 = V(r)$. The analytic continuation of $r_c(u)$ to complex $u$ is however not unique. In particular, $r_c(u)$ has branch points when other solutions to $u^2 = V(r)$ merge with $r_c(u)$ at complex $u$. These branch points were previously studied and were found to be related to the quasinormal frequencies. In particular, in the case of $l = 0$ that we restrict to, $r_c(u)$ has 4 branch points, where two correspond to the start of the lines of quasinormal frequencies in the lower half plane and the remaining correspond to their reflection into the upper half plane. The lines of quasinormal frequencies are also known to extend radially out in the complex plane. From Section 4.1.1, we know that $G(\omega_L, \omega_R)$ has poles at the quasinormal frequencies in the $\omega_L$ plane and at their reflection in the $\omega_R$ plane. Since $Z(u_L, u_R)$ depends on $r_c(u)$, one might expect branch cuts of $r_c(u)$ to remain branch cuts of $Z(u_L, u_R)$.[17] These considerations determines that the branch cut from the 4 branch points of $r_c(u)$ should extend radially outward to infinity.

Fixing the branch cuts determines the analytic continuation of $r_c(u)$ uniquely. In particular, $r_c(u)$ is analytic in a neighborhood of the origin. Continuing to purely imaginary $u$ through the region near the origin, one finds that $r_c(u)$ goes behind the horizon and approach 0 as $u \to \pm i\infty$. We also note that at small $u$, $r_c(u)$ is close to the horizon $r_0$ and we have

$$r_c(u) - r_0 \approx \frac{\beta}{4\pi} u^2 \,. \tag{66}$$

So when $u$ rotates in the complex plane by an angle $\theta$, $r_c(u) - r_0$ rotates by $2\theta$. This fact will be useful when considering the contour prescription for $Z_\psi(u)$ and $Z_\delta(u)$.

**Analytic continuation of $Z_\psi(u)$**

Now that the analytic continuation of $r_c(u)$ is fixed, we can consider the contours defining $Z_\psi$ and $Z_\delta$ in $Z$. We first consider $Z_\psi$ whose analytic continuation was worked out in [10], but we briefly review the prescription here for completeness.

$$Z_\psi(u) = \lim_{r \to \infty} \left( \log r - \int_{r_c(u)}^{r} \frac{dr'}{f(r')} \sqrt{f(r') - u^2} \right) . \tag{67}$$

The integrand has a pole at $r_0$ due to the $1/f(r)$ factor. When $u^2 > 0$, the integration contour is simply a straight line from $r_c(u)$ to infinity. For general complex $u$, one must specify the integration contour around the pole. It suffices to consider small $u$, since that is when $r_c(u)$ is close to the pole. The case of larger $u$ can simply be obtained by smoothly deforming the contour from the small $u$ case. To guarantee the function is an analytic continuation from $u > 0$, as $r_c(u)$ is analytically continued away from $u > 0$, the contour must extend smoothly from the original contour at $u > 0$ to $r_c(u)$ as in Figure 4. Notice that while $r_c(\pm i|u|)$ have the same value, $Z_\psi(\pm i|u|)$ have different contours since they are reached by rotating through different half planes. Following this prescription, as $u$ rotates by $\pm\pi$, i.e. to $u < 0$ through the upper/lower half planes, one gets two distinct contours for $Z_\psi(|u|e^{\pm i\pi})$. Remarkably, the two

---

[17]This will turn out to not be true. As mentioned, $G(\omega_L, \omega_R)$ in any one of the frequency planes only has 2 lines of quasinormal poles, but $r_c(u)$ has 4 branch cuts. It turns out the individual $Z_\psi$ and $Z_\delta$ have all 4 branch cuts of $r_c$, but they add up in a way that some cuts cancel and the final result is in agreement with $G(\omega_L, \omega_R)$.

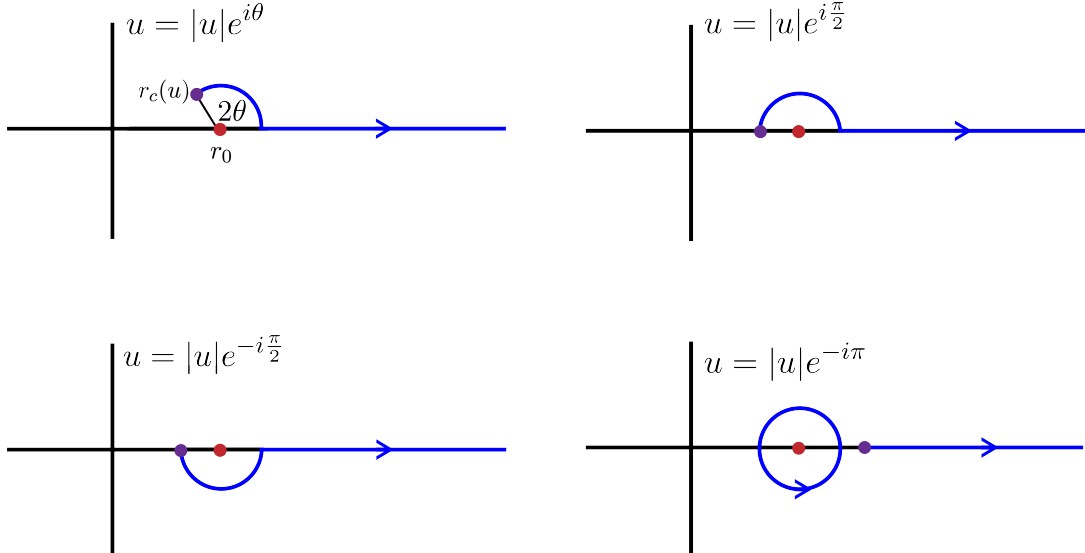

Figure 4: Top left: the contour defining $Z_\psi(u)$ for $u = |u|e^{i\theta}$ where $-\pi \leq \theta \leq \pi$. Top right: the contour first goes above the pole if $u$ is in the upper half plane. Bottom left: the contour first goes below the pole if $u$ is in the lower half plane. Bottom right: As $u$ is rotated to $-u$ through the lower half plane, the contour encloses the pole in the counterclockwise direction. If we went through the upper half plane, the contour would go clockwise instead, but the two ways of getting to $-u$ end up giving the same value. So $Z_\psi(u)$ is single-valued and analytic near $u = 0$.

contours, revolving around the pole in opposite directions, give the same contribution due to the branch cut of the square root[18] (here we use the principal branch of the square root), i.e.

$$Z_\psi(|u|e^{+i\pi}) - Z_\psi(|u|) = Z_\psi(|u|e^{-i\pi}) - Z_\psi(|u|) = |u|\frac{\beta}{2}. \tag{68}$$

Thus, we have defined $Z_\psi(u)$ in the entire complex plane using only smoothness from $u > 0$.[19] In particular, we see from the above argument that it is single-valued, which implies that it is analytic at $u = 0$. Because $r_c(u)$ appears explicitly, $Z_\psi(u)$ inherits the branch cuts of $r_c(u)$, but it is analytic everywhere else.

**Analytic continuation of $Z_\delta(u)$**

Now we turn to

$$Z_\delta(u) = \lim_{r \to r_0} \left( i \int_r^{r_c(u)} \frac{dr'}{f(r')} \sqrt{u^2 - f(r')} - iu \int_r^\infty \frac{dr'}{f(r')} \right), \tag{69}$$

and again we restrict to small $u$. At general complex $u$, if we take both integrals to be straight lines connecting the bounds, they will not approach $r_0$ in the same direction since $r_c$ is complex and the limit will not be defined. We must deform either one of the contours to line up with the other as they approach $r_0$ as in Figure 5. The precise way this is done is not important since all equivalent deformations give the same value, and we will use different equivalent prescriptions in different settings.

---

[18]Throughout the paper, we use the convention that the principal branch of a function has its branch cut in the negative real axis and takes an input $z$ with $-\pi < \arg(z) < \pi$.

[19]$Z_\psi(0)$ is defined by the limit as $u \to 0$.

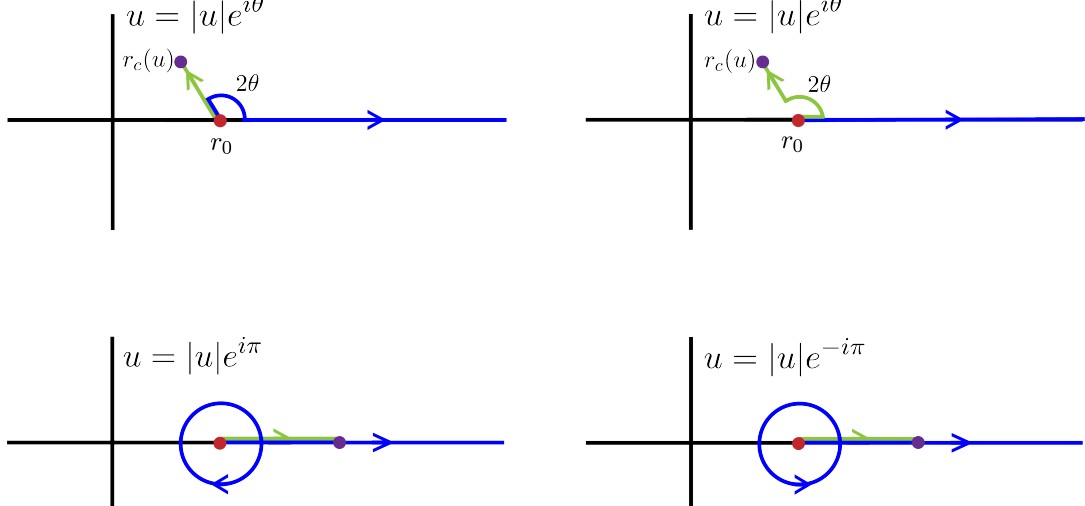

Figure 5: The green and blue contours represent the first and second integral in (70). At complex $u$, $Z_\delta(u)$ is defined by having the two integrals approach $r_0$ from the same direction. The prescriptions indicated in the top two panels have different contours but give the same result. The bottom panels adopt the prescription of the top left panel and show that one obtains a different contour if $u < 0$ is smoothly related to $u > 0$ through different half planes. Unlike $Z_\psi(u)$, these contours give different results for $Z_\delta(u)$ and one always gets a branch cut.

Aside from specifying a contour, one also needs to pick a branch of the square root to guarantee that the cancellation between the two integrals occur. To see this, we take the $r_0$-to-$r_c$ integral to be the one that is deformed, such that the integrals always approach $r_0$ from the real line. Then, writing $u = |u|e^{i\theta}$, we single out the potentially divergent parts along the last stretch of the contour (of length $\epsilon$)

$$Z_\delta(|u|e^{i\theta}) \approx i|u|\sqrt{e^{i2\theta}} \int_{r_0}^{r_0+\epsilon} \frac{dr'}{f(r')} - i|u|e^{i\theta} \int_{r_0}^{r_0+\epsilon} \frac{dr'}{f(r')} + \text{finite}, \tag{70}$$

and we see that the divergent parts cancel if we take $\sqrt{e^{i2\theta}} = e^{i\theta}$ for the angles that are smoothly deformed from $u > 0$. This guarantees that the quantity is well-defined and the contour ensures that the function is analytic at least in a neighbourhood of $u > 0$. So for general $\theta$, the integral is given by

$$Z_\delta(|u|e^{i\theta}) = i\sqrt{e^{i2\theta}} \int_{r_0}^{r_c(|u|e^{i\theta})} \frac{dr'}{f(r')} \sqrt{|u|^2 - \frac{f(r')}{e^{-i2\theta}}} - i|u|e^{i\theta} \int_{r_0}^{\infty} \frac{dr'}{f(r')}, \tag{71}$$

where the square root inside the integrand is the principal branch and the one in the prefactor is defined by what angles we choose to be smoothly deformed from $u > 0$.

The above choice of branches of square root suggests that $Z_\delta(u)$ is not single-valued. It is easier to see this by taking the $r_0$-to-$r_c$ integral to go along the straight contour and have the integral contour defining $r_*$ be deformed. Since the $r_*$ integral does not have a square root, it is easy to see that it cannot give the same value at $u < 0$.

$$Z_\delta(|u|e^{+i\pi}) - Z_\delta(|u|) \neq Z_\delta(|u|e^{-i\pi}) - Z_\delta(|u|). \tag{72}$$

Thus, $Z_\delta(u)$ is not single-valued and $u = 0$ is a branch point. In the calculation of (72), the branch cut is placed at $u < 0$, but this need not be the case and, in fact, just with the integral

alone there is no natural way to choose a branch cut. Here the analyticity properties discussed in Appendix A.2 become crucial since it determines where the branch cut must be placed. Before taking the large mass limit, $e^{-i\delta_{\omega_R}}C(\omega_R)$ has a line of zeroes along the positive imaginary axis that cancel the poles from the gamma functions. In the large mass approximation, we consider the logarithm of $e^{-i\delta_{\omega_R}}C(\omega_R)$, and the line of zeroes become a branch cut in $Z_\delta(u_R)$, which is to be canceled by another branch cut from the Stirling's approximation of the gamma functions. This means that for $Z_\delta(u_R)$ we should place the cut in the positive imaginary $u_R$ axis, whereas for $Z_\delta(u_L)$, we need to choose the branch cut differently according to the analyticity of $e^{i\delta_{\omega_L}}C(\omega_L)$, which indicates the cut is to be placed in the negative imaginary $u_L$ axis.

**Analytic continuation of $Z(u_L, u_R)$**

Finally we now put every term together to get the full $Z(u_L, u_R)$ in the complex plane. There are two noteworthy features. One is that while both $Z_\psi(u)$ and $Z_\delta(u)$ inherit all four branch cuts of $r_c(u)$, when they are combined into $Z_\psi(u_L)+Z_\delta(u_L)$ and $Z_\psi(u_R)-Z_\delta(u_R)$, some branch cuts will be cancelled. To see this, consider $Z_\psi(u_R)-Z_\delta(u_R)$ with $u_R = |u_R|e^{i\theta}$.

$$
\begin{aligned}
Z_\psi(|u_R|e^{i\theta}) - Z_\delta(|u_R|e^{i\theta}) &= \log r_\infty - \int_{r_c(|u|e^{i\theta})}^{r_\infty} \frac{dr'}{f(r')}\sqrt{f(r')-|u|^2 e^{i2\theta}} \\
&\quad - i\sqrt{e^{i2\theta}}\int_{r_0}^{r_c(|u|e^{i\theta})} \frac{dr'}{f(r')}\sqrt{|u|^2 - \frac{f(r')}{e^{-i2\theta}}} + i|u|e^{i\theta}\int_{r_0}^{\infty}\frac{dr'}{f(r')},
\end{aligned}
\tag{73}
$$

where $r_\infty$ is to be taken to $\infty$. As mentioned, we take the branch cut from $u_R = 0$ to go in the positive imaginary axis, so the branch of square root in the prefactor of the third term is $\sqrt{e^{i2\theta}} = e^{i\theta}$ for $\pi/2 > \theta > -3\pi/2$. We will write the integrand of the third term in the same form as the integrand of the second term, whose square root is in the principal branch. This requires absorbing $\pm i$ of the prefactor $i\sqrt{e^{i2\theta}}$ back into a principal branch of square root. We get $i\sqrt{e^{i2\theta}} = -\sqrt{-e^{i2\theta}}$ for $e^{i\theta}$ in the upper half plane and $i\sqrt{e^{i2\theta}} = \sqrt{-e^{i2\theta}}$ for $e^{i\theta}$ in the lower half plane.[20] So for $u_R$ in the upper half plane, we have

$$
\begin{aligned}
Z_\psi(|u_R|e^{i\theta}) - Z_\delta(|u_R|e^{i\theta}) &= \log r_\infty - \int_{r_c(|u|e^{i\theta})}^{r_\infty} \frac{dr'}{f(r')}\sqrt{f(r')-|u|^2 e^{i2\theta}} \\
&\quad - \int_{r_c(|u|e^{i\theta})}^{r_0} \frac{dr'}{f(r')}\sqrt{f(r')-|u|^2 e^{i2\theta}} + i|u|e^{i\theta}\int_{r_0}^{\infty}\frac{dr'}{f(r')},
\end{aligned}
\tag{74}
$$

where the $r_c(u_R)$ dependence remains, whereas $r_c(u_R)$ does not appear when $u_R$ is is the lower half plane

$$
Z_\psi(|u_R|e^{i\theta}) - Z_\delta(|u_R|e^{i\theta}) = \log r_\infty - \int_{r_0}^{r_\infty}\frac{dr'}{f(r')}\sqrt{f(r')-|u|^2 e^{i2\theta}} + i|u|e^{i\theta}\int_{r_0}^{\infty}\frac{dr'}{f(r')}.
\tag{75}
$$

This is to say, of the four branch cuts of $r_c(u)$, the ones in the lower half plane no longer appear for $Z_\psi(u_R)-Z_\delta(u_R)$, but the ones in the upper half plan remain. This takes $Z(u_L, u_R)$ closer to the analytic structure of $G(\omega_L, \omega_R)$ in the $\omega_R$ plane, which has lines of poles at the reflection of

---

[20]We want to write $i\sqrt{e^{i2\theta}}$ in the specific choice of square root mentioned into $\pm\sqrt{e^{i(2\theta+\alpha)}}$ in the principal branch, where $2\theta + \alpha$ is within $(-\pi, \pi)$. Therefore, for $0 < \theta < \frac{\pi}{2}$, we absorb $-i = e^{-i\frac{\pi}{2}}$ so that $i\sqrt{e^{i2\theta}} = -\sqrt{e^{i(2\theta-\pi)}}$; while for $-\frac{3}{2}\pi < \theta < -\pi$, we absorb $-i = e^{i\frac{3\pi}{2}}$ to get $i\sqrt{e^{i2\theta}} = -\sqrt{e^{i(2\theta+3\pi)}}$; and for $-\pi < \theta < \pi$, we take $i = e^{i\frac{\pi}{2}}$ to get $i\sqrt{e^{i2\theta}} = \sqrt{e^{i(2\theta+\pi)}}$.

the quasinormal frequencies into the upper half plane but not at the quasinormal frequencies themselves.

This brings us to the second feature of $Z(u_L, u_R)$. $G(\omega_L, \omega_R)$ in the $\omega_R$ plane only has one more line of poles, starting from $\omega_L$ extending into the lower half plane. This is represented by $\log(u_L - u_R)$ in $Z_{matching}$, where the branch cut of the logarithm in the $u_R$ plane is chosen to point down from $u_L$. From the general analysis in 4.1.1, $G(\omega_L, \omega_R)$ does not have further poles and it does not have zeroes, but $Z(u_L, u_R)$ has two terms with a branch cut in the $u_R$ plane starting from 0 extending in the positive imaginary axis: $i\frac{\beta}{2\pi} u_R \log(\alpha u_R/(u_L - u_R))$ in $Z_{matching}$, and $-Z_\delta(u_R)$. As mentioned before, the two contributions cancel each other in accordance with the analytic structure of $G(\omega_L, \omega_R)$. Since neither functions are divergent at the branch cut, the branch cuts cancel out if the sum is single-valued, which can be checked easily by a calculation similar to (72). Thus, $Z(u_L, u_R)$ in the $u_R$ plane has analytic properties that are in agreement with those of $G(\omega_L, \omega_R)$ in the $\omega_R$ plane. By repeating the same analysis, one can also check the same for $\omega_L$. In this way, we have a complete analytic continuation of $Z(u_L, u_R)$.

### 4.3.2 Relation to geodesics

Through the above discussion, we see that $Z(u_L, u_R)$ is well-defined in a certain range of imaginary frequencies. We will now show that at imaginary frequencies, $Z(u_L, u_R)$ is closely related to spacelike geodesics that pass through the interior.

It is useful to define the following integrals between $r = a$ and $r = b$

$$
\begin{aligned}
T_\pm(a, b; E) &\equiv \pm \int_a^b \frac{dr}{f(r)} \frac{E}{\sqrt{E^2 + f(r)}}, \\
L(a, b; E) &\equiv \int_a^b \frac{dr}{\sqrt{E^2 + f(r)}}, \\
I(a, b; E) &\equiv \int_a^b \frac{dr}{f(r)} \sqrt{E^2 + f(r)} = ET_+(a, b; E) + L(a, b; E).
\end{aligned}
\tag{76}
$$

Note that $I$ is the Legendre transform of $L$ since

$$
\frac{\partial I(a, b; E)}{\partial E} = T_+(a, b; E).
\tag{77}
$$

This holds both in the case where $a, b$ are independent of $E$ and when the endpoints $a, b$ are turning points at $E$.

These integrals can be interpreted physically in terms of geodesics. When we take $b > a$, $L(a, b; E)$ is the proper length and $T_+(a, b; E)$ is the Schwarzschild time difference between the points $r = a$ and $r = b$ along a geodesic with energy $E$ assuming $r$ is monotonic along the geodesic. Since the geodesic is spacelike, $T_+(a, b; E)$ could be either $t_b - t_a$ or $t_a - t_b$ depending on the situation. The time difference is more complicated if $r$ is not monotonic along the geodesic. Consider a geodesic with energy $E$ starting at $r = a$ and ending at $r = b$ while going through a turning point $r = c$ (we can assume $a > c$ and $b > c$). In this case, it is more convenient to parameterize the geodesic by the proper length $\lambda$ increasing from $\lambda_a$ at the starting point to $\lambda_b$ at the endpoint. The Schwarzschild time difference (up to a sign

depending on the geodesic) is then

$$
\begin{aligned}
[t_b - t_a]_E &= \int_{\lambda_a}^{\lambda_b} d\lambda \frac{dt}{d\lambda} \\
&= \int_{\lambda_a}^{\lambda_c} d\lambda \frac{E}{f} + \int_{\lambda_c}^{\lambda_b} d\lambda \frac{E}{f} \\
&= -\int_a^c dr \frac{E}{f(r)\sqrt{f(r)+E^2}} + \int_c^b dr \frac{E}{f(r)\sqrt{f(r)+E^2}} \\
&= T_+(c,a;E) + T_+(c,b;E) = T_-(a,c) + T_+(c,b),
\end{aligned}
\tag{78}
$$

where we use the notation $[t_b - t_a]_E$ to denote the fact that we will be thinking of the Schwarzschild time difference as a function of only $E$, the energy of the geodesic. Note that in the third line we used the fact that $r(\lambda)$ is not monotonic. The notation in the last line indicates that it is useful to write the coordinates $r$ from left to right in the direction of increasing $\lambda$ and using $T_-$ for segments where $\lambda$ increases in the direction of decreasing $r$. This will be useful for interpreting $Z(u_L, u_R)$.

Now we see that integrals of the form $I(a, b; iu)$ appear in (62). As mentioned, rotating from $u > 0$ to $u = -iE$ for real $E$ should make the connection to geodesics manifest. It is useful to first exclude $Z_{matching}$ and look at the other terms, and we will consider the case of $E_L, E_R > 0$.[21] For $a = L, R$,

$$
Z_\psi(-iE_a) = \lim_{r\to\infty} \left( \log r - \int_{r_c(-iE_a)}^r \frac{dr'}{f(r')} \sqrt{f(r')+E_a^2} \right),
\tag{79}
$$

$$
Z_\delta(-iE_a) = \lim_{r\to r_0} \left( i \int_r^{r_c(-iE_a)} \frac{dr'}{f(r')} \sqrt{-E_a^2 - f(r')} - E_a \int_r^\infty \frac{dr'}{f(r')} \right)
\tag{80}
$$

$$
= \lim_{r\to r_0} \left( \int_r^{r_c(-iE_a)} \frac{dr'}{f(r')} \sqrt{E_a^2 + f(r')} - E_a \int_r^\infty \frac{dr'}{f(r')} \right).
\tag{81}
$$

In (81), we chose $\sqrt{-1} = -i$ in accordance with (71). When we combine the terms in $Z$ after taking $u = -iE$, we see that the portion of the $u_R$ integral from $r_c(-iE_R)$ to $r_0$ cancels, while the corresponding terms of the $u_L$ integral do not cancel and double instead (a special case of (74) and (75)), so we have

$$
\begin{aligned}
Z &= \lim_{\substack{r_\infty \to \infty \\ r\to r_0}} \left[ 2\log(r_\infty) - \int_r^{r_\infty} \frac{dr'}{f(r')} \sqrt{f(r')+E_R^2} + E_R \int_r^\infty \frac{dr'}{f(r')} \right. \\
&\qquad \left. - \left( \int_{r_c(-iE_L)}^{r_\infty} \frac{dr'}{f(r')} \sqrt{f(r')+E_L^2} + \int_{r_c(-iE_L)}^r \frac{dr'}{f(r')} \sqrt{f(r')+E_L^2} \right) - E_L \int_r^\infty \frac{dr'}{f(r')} \right] \\
&\quad + Z_{matching}(-iE_L, -iE_R) \\
&= -E_R[r_*(r_0) + T_+(r_0, \infty; E_R)] - L_{reg}(r_0, \infty; E_R) \\
&\quad - E_L[T_-(\infty, r_c(-iE_L); E_L) + T_+(r_c(-iE_L), r_0; E_L) - r_*(r_0)] \\
&\quad - \big[ L_{reg}(r_c(-iE_L), \infty; E_L) + L(r_c(-iE_L), r_0; E_L) \big] \\
&\quad + Z_{matching}(-iE_L, -iE_R).
\end{aligned}
\tag{82}
$$

$$
\tag{83}
$$

---

[21]Excluding $Z_{matching}$ means we will be considering functions with branch cuts at $E_R < 0$ and $E_L > 0$, so taking $u_L = -iE_L$ here means $u_L = \lim_{\epsilon\to 0^+} E_L e^{i(-\pi/2+\epsilon)}$. Once we consider all the terms in $Z(u_L, u_R)$, there is no ambiguity at $u_L = -iE_L$.

Instead of writing the limits explicitly, we have defined $L_{reg}$ by absorbing the $\log(r_\infty)$ such that evaluating $L_{reg}$ at $\infty$ is equivalent to taking $r_\infty \to \infty$ in (82), and we have abused notation by abbreviating the $r \to r_0$ limit of (82) by evaluating $T$ and $r_*$ at $r_0$ even though it is only the specific combination that is well-defined in the limit. Note that using our contour prescription, the second line of (83) acquires an imaginary part $-i\beta E_L/2$.

Eq. (83) can be interpreted geometrically. The first line contains terms that depend only on $E_R$. The quantity in brackets is, roughly speaking, in the form of a time difference of a geodesic going from the horizon to the boundary without a turning point. Of course the Schwarzschild time at the horizon is not defined, but it appears together with the tortoise coordinate $r_*$ to give the well-defined Eddington-Finkelstein coordinate $v$ at the horizon. Since these terms depend on $E_R$, we can interpret them as a quantity related to a geodesic in the right exterior. As we shall shortly see, the $E_L$ terms are in a form that can be related to a geodesic with a turning point. If these geodesics are to be joined together (and one does get a natural interpretation for all terms in $Z$ if one assumes this), it must be that the right geodesic hits the future horizon (i.e. the shock wave) rather than the past horizon. As the geodesic goes from the right boundary toward the future horizon, the Schwarzschild time increases, so $T_+(r_0, \infty; E_R)$ must be interpreted as the time at the horizon minus the time at infinity. So the quantity in brackets can be denoted by

$$[v_R - t|_{\partial_R}]_{E_R} \equiv r_*(r_0) + T_+(r_0, \infty; E_R), \tag{84}$$

which represents a function purely of $E_R$ but can be thought of as the difference between the (right) Eddington-Finkelstein $v_R$ at the shock wave and the bulk $t$ coordinate at the right boundary for a geodesic at energy $E_R$.[22] Now that we have related the terms in the brackets to quantities associated to a geodesic in the right exterior, it is easy to see that the remaining term in the first line of (83), $L_{reg}(r_0, \infty; E_R)$, is exactly the regularized length of that geodesic.

Next we turn to terms in the second and third line of (83), which depend only on $E_L$. As mentioned, $r_c(-iE_L)$ is behind the horizon, so we can interpret the second line using (78) with $r_c(-iE_L)$ being the turning point of a geodesic starting from the horizon $r = r_0$, going through the interior, and ending at the left boundary at $r = \infty$. Since this geodesic goes through the interior and is associated to the left, $r = r_0$ here must represent the right future horizon. We use (78) to again interpret the quantities in the brackets as a difference between a Schwarzschild time and an Eddington-Finkelstein coordinate. Note that $f$ is negative in the integral defining $T_+(r_c(-iE_L), r_0; E_L)$. So $T_+(r_c(-iE_L), r_0; E_L)$ is negative and corresponds to $[t_{r_c} - t_{r_0}]_{E_L}$. This determines that the quantities in the brackets can be interpreted as

$$[t|_{\partial_L} - v_L]_{E_L} \equiv T_-(\infty, r_c(-iE_L); E_L) + T_+(r_c(-iE_L), r_0; E_L) - r_*(r_0). \tag{85}$$

It is then also clear that the third line is (minus) the total proper length of the geodesic starting from the shock wave, passing through the interior and ending at the left boundary. We denote that as

$$\bar{L}_{reg}(r_0, \infty; E_L) \equiv L_{reg}(r_c(-iE_L), \infty; E_L) + L(r_c(-iE_L), r_0; E_L), \tag{86}$$

where the bar signifies that the geodesic passes through the interior.

We can then write

$$Z = -E_L[t|_{\partial_L} - v_L]_{E_L} + Z_{matching} - E_R[v_R - t|_{\partial_R}]_{E_R} - \bar{L}_{reg}(r_0, \infty; E_L) - L_{reg}(r_0, \infty; E_R), \tag{87}$$

where except for $Z_{matching}$, every term appearing in $Z$ has an interpretation in terms of quantities associated to two piecewise geodesics starting from opposite boundaries and ending at

---

[22]Note that the quantities inside the brackets are not individually specified (in particular, $t|_{\partial_R}$ is not a given time since we are in frequency space) and must be understood as differences that are functions of $E_R$. We will use the same notation for quantities pertaining to $E_L$.

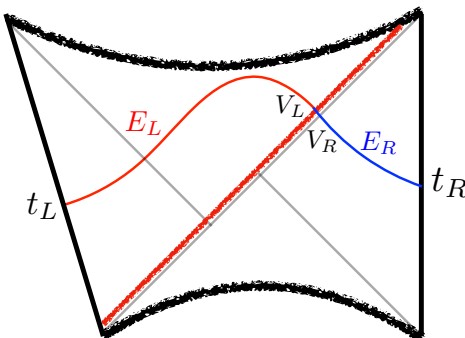

Figure 6: At imaginary frequencies, terms appearing in $Z$ have interpretations in terms of quantities associated to two pieces of geodesics between the boundaries and the shock wave.

the shock wave (see Figure 6). However, if these piecewise geodesics (labeled by energy $E_a$) are to link up at the shock wave appropriately to form a geodesic across the entire space time, the place at which they meet at the shock wave, labeled by the Kruskal coordinate on the corresponding patches, must satisfy the condition

$$\frac{E_L}{E_R} = \frac{V_L}{V_R} = \frac{V_R + \alpha}{V_R} = \frac{V_L}{V_L - \alpha}. \tag{88}$$

Note that this implies that for $\alpha > 0$, which is what we are assuming, there are no radial geodesics with $E_R > E_L$ connecting the two boundaries. One can solve (88) to find the Eddington-Finkelstein coordinates at which the geodesic hits the shock wave from each side as functions of $E_L$ and $E_R$

$$[v_R]_{E_L,E_R} \equiv \frac{\beta}{2\pi} \log \frac{\alpha E_R}{E_L - E_R}, \tag{89}$$

$$[v_L]_{E_L,E_R} \equiv \frac{\beta}{2\pi} \log \frac{\alpha E_L}{E_L - E_R}. \tag{90}$$

One then notices that these naturally occur within $Z_{matching}(-iE_L, -iE_R)$

$$Z_{matching}(-iE_L, -iE_R) = E_R \frac{\beta}{2\pi} \log\left(\frac{\alpha E_R}{E_L - E_R}\right) + E_L \frac{\beta}{2\pi} \log\left(\frac{E_L - E_R}{\alpha E_L}\right) + i\frac{\beta}{2} E_L \tag{91}$$

$$= E_R[v_R]_{E_L,E_R} - E_L[v_L]_{E_L,E_R} + i\frac{\beta}{2} E_L. \tag{92}$$

Therefore, we see that

$$Z(-iE_L, -iE_R) = -E_L\big([t|_{\partial_L} - v_L]_{E_L} + [v_L]_{E_L,E_R}\big) - E_R\big(-[v_R]_{E_L,E_R} + [v_R - t|_{\partial_R}]_{E_R}\big)$$
$$- \bar{L}_{reg}(r_0, \infty; E_L) - L_{reg}(r_0, \infty; E_R) + i\frac{\beta}{2} E_L, \tag{93}$$

where the $+i\frac{\beta}{2} E_L$ cancels exactly with the imaginary part coming from our contour prescription in computing $-E_L[t|_{\partial_L} - v_L]_{E_L}$.

The above expression can be understood in an intuitive way. Suppose we Fourier transform $\omega_R, \omega_L$ back to position space $t_R, t_L$ and evaluate the integral by the method of steepest descent, we would need the saddle points of $Z(u_L, u_R) - iu_R t_R - iu_L t_L$. We can look for solutions at real $E = iu$ for $E > 0$ by using the expression in (93) for $Z(u_L, u_R)$. The saddles are given

by $\partial_{E_a}[Z(-iE_L, -iE_R) - E_R t_R - E_L t_L] = 0$, which evaluate to

$$t_R = [t|_{\partial_R} - v_R]_{E_R} + [v_R]_{E_L, E_R}, \tag{94}$$

$$-t_L = [t|_{\partial_L} - v_L]_{E_L} + [v_L]_{E_L, E_R} - i\frac{\beta}{2}, \tag{95}$$

where in this notation $t_{L/R}$ on the left are specific boundary times, whereas the expressions on the right are purely a function of energies. Note that in (95) $t_L$ and $t|_{\partial_L}$ have opposite sign because $t_L$ is the future pointing boundary time and $t|_{\partial_L}$ is the left bulk time which increases to the past, and $-i\beta/2$ is canceled by an imaginary part from $[t|_{\partial_L} - v_L]_{E_L}$. Equations (94) and (95) suggest that at the saddle, the Eddington-Finkelstein coordinates at which the geodesic on one side hits the shock wave are exactly those of a geodesic on the entire spacetime. This shows that saddles of $Z$ satisfy the geodesic equations, but that does not imply that the position space two-point function is dominated by real geodesics. In fact, we expect the bouncing geodesic that was mentioned in Section 2 to not dominate the real time two-point function based on the case without the shock wave [9,10]. Instead, the real time two-point function is expected to be dominated by complex solutions to the geodesic equations (94), (95)[23] that do not have a real spacetime geometric interpretation.

## 4.4 Probing the singularities via saddles

In Section 2 we defined $t_L(t_R)$ (13) to be the latest time that an observer could jump into the black hole from the left and receive a signal sent from the right boundary at time $t_R$. In this section, we show how to recover $t_L(t_R)$ from the two-point function. The key is to use the relationship to geodesics at imaginary frequencies. Since we want to detect the singularity bending down as a function of $t_R$, we keep $u_L = -iE_L$ imaginary and Fourier integrate $u_R$ to $t_R$ via the method of stationary phase.

$$G(\omega_L, t_R) = \int \frac{d\omega_R}{2\pi} e^{-i\omega_R t_R} G(\omega_L, \omega_R) \approx \frac{\nu}{2\pi} \int du_R \frac{2\nu^{1/2}\beta^{1/2}}{\sqrt{i(u_L - u_R)}} e^{\nu(-iu_R t_R + Z(u_L, u_R))}. \tag{96}$$

We will show that for large enough $E_L$, the saddle point dominating the integral is given by contributions from the bouncing geodesic and one will be able to detect the effect of the shock wave on the bending down of the singularity. This is manifested in the exponential behavior of the mixed frequency-time correlator (96) as $E_L$ grows.

It is important that in (96) we only Fourier transform back one of the frequencies as the real time two-point function is not expected to be dominated by the real geodesic saddle. The constraint that the left frequency stays imaginary is also crucial for picking out the real geodesic. The requirement that $E_L > 0$ is related to the fact that there is a branch cut for $E_R > E_L$: for $E_L > 0$, the branch cut does not intersect the real line in the $\omega_R$ plane, so the $\omega_R$-contour defining $G(\omega_L, t_R)$ can be simply taken to be the real line; if we considered $E_L < 0$, the branch cut goes through the real line and the $\omega_R$-contour must undergo a large deformation into the lower half plane for $G(\omega_L, t_R)$ to even be defined, but the resulting quantity does not have a clear physical meaning.

Our discussion so far applies to general dimensions. In this section, we will restrict to $D = 5$, where we are able to explicitly evaluate the integrals for $Z$ in order to calculate the Fourier transform (96). However, we expect the same features to hold in any dimensions $D > 3$. The case of $D = 5$ is easier to work with because $f(r)$ factorizes

$$f(r) = \frac{(r^2 - r_0^2)(r^2 + r_1^2)}{r^2}, \tag{97}$$

---

[23]Although these equations are derived for $E_L, E_R > 0$, they also hold for complex $E_a$ in a neighbourhood of $E_L, E_R > 0$. Further away from this region, i.e. for $u_a$ in different half planes, the equations are still geodesic equations but are not given by (93).

where $r_1^2 = r_0^2 + 1$ and $\mu = r_0^2 r_1^2$. In terms of $r_0, r_1$, we have

$$\beta = \frac{2\pi r_0}{r_0^2 + r_1^2}, \qquad \tilde{\beta} = \frac{2\pi r_1}{r_0^2 + r_1^2}. \tag{98}$$

The branch points $u_i$ of $r_c(u)$ also take a simple form

$$u_1 = ir_0 + r_1, \quad u_2 = ir_0 - r_1, \quad u_3 = u_2^* = -ir_0 - r_1, \quad u_4 = u_1^* = -ir_0 + r_1. \tag{99}$$

Defining $C_i^a = u_a - u_i$ for $a = L, R$ and $i = 1, \dots, 4$, we can write $Z(u_L, u_R)$ as

$$
\begin{aligned}
Z(u_L, u_R) = {} & \frac{1}{2} \log\left( \frac{C_1^R C_2^R C_3^L C_4^L}{16} \right) + \frac{iu_R \beta}{4\pi} \left( \log C_1^R C_2^R - 2\log\left( i\frac{u_L - u_R}{\alpha} \right) \right) \\
& + \frac{iu_L \beta}{4\pi} \left( \log \frac{1}{C_3^L C_4^L} + 2\log\left( i\frac{u_L - u_R}{\alpha} \right) \right) - (u_L + u_R)\frac{\beta}{4} \\
& + \frac{u_R \tilde{\beta}}{4\pi} \log \frac{C_2^R}{C_1^R} + \frac{u_L \tilde{\beta}}{4\pi} \log \frac{C_3^L}{C_4^L},
\end{aligned}
\tag{100}
$$

where there is a branch point associated with every $C_i^a$ in the $u_a$ plane at $u_i$ and the branch cut is chosen to extend to infinity from $u_i$.[24] The branch cut associated with $u_L - u_R$ is chosen to be at $\mathrm{Im}(u_L - u_R) < 0$, $\mathrm{Re}(u_L) = \mathrm{Re}(u_R) = 0$. The analytic structure is therefore in agreement with what was discussed in Section 4.3.1.

With the explicit expression (100), we now evaluate the Fourier integral (96) via the method of steepest descent. This involves first finding the saddles of $-iu_R t_R + Z(u_L, u_R)$ with respect to $u_R$ and then finding the steepest descent contours. The saddles are solutions to

$$t_R = i\frac{\beta}{4} - i\frac{\tilde{\beta}}{4\pi} \log \frac{C_2^R}{C_1^R} + \frac{\beta}{4\pi} \left( \log C_1^R C_2^R - 2\log\left( i\frac{u_L - u_R}{\alpha} \right) \right), \tag{103}$$

where the logarithms are again in the same branches as in (100). Switching to $E_a = iu_a$ for $a = L, R$, the equation becomes[25]

$$t_R + \frac{\tilde{\beta}}{2\pi} \left( \arctan \frac{E_R + r_0}{r_1} - \frac{\pi}{2} \right) + \frac{\beta}{4\pi} \log \frac{(E_L - E_R)^2}{\alpha^2 \left( (E_R + r_0)^2 + r_1^2 \right)} = 0. \tag{104}$$

In general, the solutions to this equation will be complex and do not correspond to real space-time geodesics, but it turns out that at fixed $t_R$, if we take $E_L$ to be large enough, then the solutions will be at real $E_R$.

To see how this is the case, we restrict attention to real $E_R$. Notice that the imaginary part of the left hand side of (104) vanish identically. Furthermore, one should only consider this expression for $E_L > E_R$ because of the branch cut, which reflects the fact that there are

---

[24]More explicitly, our choice of branch here means

$$\log\left( \Pi_i (u - u_i)^{a_i} \right) = \sum_i a_i \log(u - u_i), \tag{101}$$

where $\log(u - u_i)$ has a branch cut extending from $u_i$ to infinity radially and the expression agrees with the principal branch of the logarithm in a neighbourhood of $u > 0$. For example, for $u_i$ in the upper half plane and let $\theta = \arg(u_i)$ in the principal branch, we have

$$\log(u - u_i) = \log|u - u_i| + i\left[ \arg\left( (u - u_i)e^{-i(\theta - \pi)} \right) + \theta - \pi \right]. \tag{102}$$

[25]Note that $i\beta/4$ is cancelled by evaluating $\log C_1^R C_2^R$ of (103) in accordance with the specified branch cuts (102).

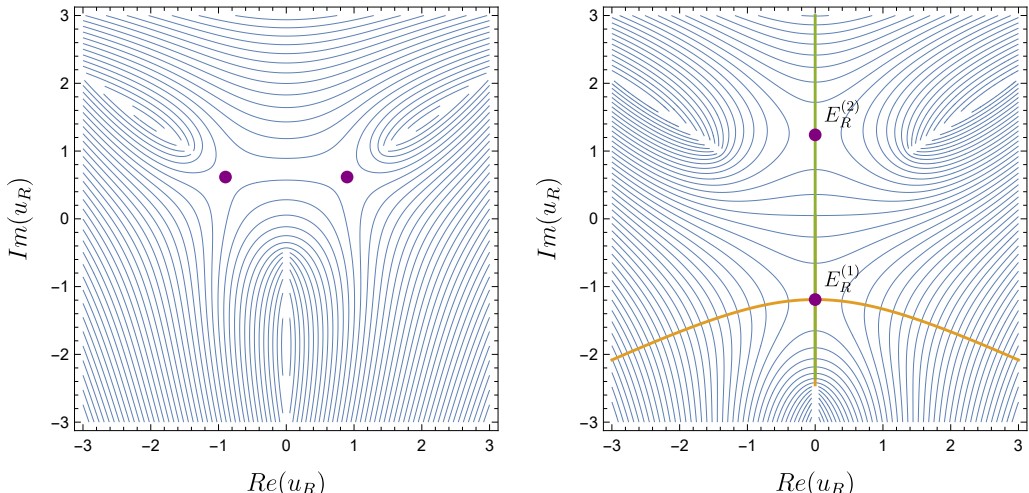

Figure 7: The contour plot of the real part of $Z(u_L, u_R) - iu_R t_R$ in the $u_R$ plane at fixed $t_R$ and different $E_L = iu_L > 0$. The two branch cuts in the upper half plane come from $r_c(u)$ and are associated with the reflection of the quasinormal poles. The branch cut in the lower half plane is the one of $Z_\delta(u_R)$ and starts at $u_L$. For smaller $E_L$ (left), the saddles are at complex $u_R$. If $E_L$ is large enough (right), the saddles lie on the imaginary axis. In the latter case, the steepest descent contours through $E_R^{(1)}$ and $E_R^{(2)}$ are the yellow and green lines respectively. The original contour can only deform into the yellow contour.[26]

no geodesics with $E_R > E_L$. We consider the expression on the left hand side of (104) as a function of $E_R$ for a fixed $E_L$. It has one maximum in the relevant region of $E_L > E_R$ when $E_L > -2\pi/\beta$ (which is always true for $E_L > 0$). Since the expression approaches $-\infty$ as $E_R \to E_L$, we always have at least one real solution if the maximum is greater than 0. This will not be true if $E_L$ is small, in which case one obtains complex saddles. But if we take $E_L$ to be large, the maximum of the left hand side can be made as big as possible due to the presence of $\log E_L$ and one is guaranteed to get real saddles.

Since $t_R$ appears also on the left hand side, how big $E_L$ needs to be depends on $t_R$, e.g. for large negative $t_R$, the minimum $E_L \sim e^{-\frac{2\pi}{\beta} t_R}$. In this regime, there is a second real solution if the function on the left hand side of (104) decreases below 0 as $E_R \to -\infty$, which occurs when

$$t_R < t_R^* = \frac{\tilde{\beta}}{2} + \frac{\beta}{2\pi} \log \alpha . \tag{105}$$

We denote the solution that always appear by $E_R^{(1)}$ and the solution that appears only when $t_R < t_R^*$ by $E_R^{(2)}$. Note that $E_R^{(1)} \geq E_R^{(2)}$.

As explained in the previous paragraph, the existence of real saddles relies on $E_L$ to be larger than a $t_R$ dependent value, but does not require $E_L \to \infty$. The limit $E_L \to \infty$ is useful because the spacelike geodesics of these saddles become increasingly null and turn into geodesics that bounces at the singularity. In this limit,

$$E_R^{(1)} \approx \frac{E_L}{1 + \alpha e^{\frac{-2\pi t_R}{\beta}}} , \tag{106}$$

which corresponds to a spacelike geodesic that bounces at the future singularity, as mentioned

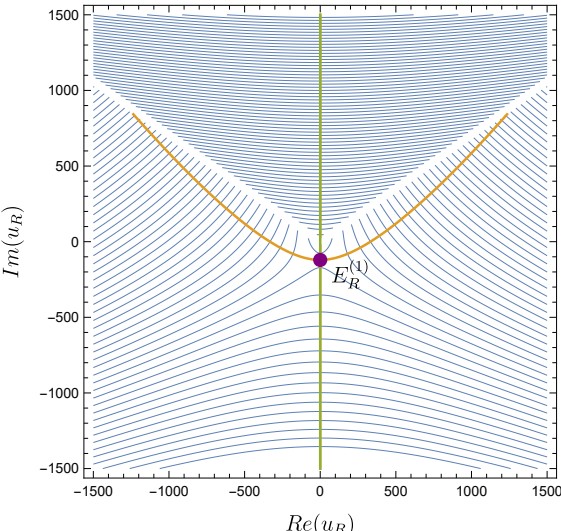

Figure 8: The steepest descent contour for $E_R^{(1)}$ becomes a curve of constant slope at large $|u_R|$. We note that while a saddle $E_R^{(2)}$ is present in the figure above, we do not denote it since our focus is on the contour passing through $E_R^{(1)}$. The slope of the steepest descent contour for $E_R^{(1)}$ is dependent on $t_R$. For example, the slope on the right asymptotically is given by $\frac{1}{\pi}\log\alpha - \frac{2t_R}{\beta}$. The case of Figure 7 has a contour that continues into the lower half plane, whereas here we have a contour that continues into the upper half plane. In either case, the contour can be deformed from the real axis by closing the contour at infinity, unlike the steepest descent contour of $E_R^{(2)}$. The steepest descent contour can run into the branch cuts in the upper half plane if the slope of the contour is large enough. In $D = 5$, this occurs when $\frac{1}{\pi}\log\alpha - \frac{2t_R}{\beta} > \frac{\beta}{\tilde{\beta}}$. It is more complicated in other dimensions since the branch cut does not have constant slope. Despite this, one can simply have the contour run along the branch cut to infinity, which would again be deformable from the real axis. In this case, the contribution of the integral along the branch cut is exponentially small compared to that of the saddle.

in Section 2.2. The other solution for $t_R < t_R^*$ is given by

$$E_R^{(2)} \approx \frac{E_L}{1 - \alpha e^{\pi\frac{\tilde{\beta}}{\beta}} e^{-\frac{2\pi t_R}{\beta}}}, \tag{107}$$

which corresponds to a geodesic that, when traced starting from the right, bounces first at the past singularity and then again at the future singularity before reaching the left boundary. There are further geodesic saddles if we take $E_L < 0$, corresponding to geodesics that bounce at the past singularity, but as mentioned these do not have a clear physical meaning.

Now that the saddles are found, one can examine the steepest descent contours that pass through each of these saddles. As shown in Figures 7 and 8, the original contour defining (96) can only deform into the contour through $E_R^{(1)}$. Therefore, the steepest descent method gives

---

[26]Each of the saddles is a local maximum on their respective steepest descent contours, but is a local minimum in the orthogonal direction. This implies that if we deform the real axis to go through $E_R^{(2)}$ in the horizontal direction, that saddle would not dominate the integral. The behaviour of $Z(u_L, u_R)$ at large $u_R$ does not allow deforming the real line to the green contour.

the following approximation for (96) at large $E_L$

$$G(-i\,\nu E_L, t_R) \approx 2\nu \left(\frac{E_L^2}{4\left(1 + \alpha e^{-\frac{2\pi t_R}{\beta}}\right)}\right)^{\nu} \exp\left[-\nu E_L\left(t_R + \frac{\tilde{\beta}}{2} + \frac{\beta}{2\pi}\log\left(1 + \alpha e^{-\frac{2\pi t_R}{\beta}}\right)\right)\right].$$
(108)

We see that $G(\omega_L, t_R)$ is dominated by an exponential function as $\omega_L \to -i\infty$. The coefficient governing the growth/decay of the exponential function

$$-t_R - \frac{\tilde{\beta}}{2} - \frac{\beta}{2\pi}\log\left(1 + \alpha e^{-\frac{2\pi t_R}{\beta}}\right),$$
(109)

is exactly $t_L(t_R)$ (13), the latest time on the left boundary at which an observer can still receive a signal entering the black hole from the right at $t_R$.

## 5 Discussion

We have considered the thermofield double state with a perturbation on the left at time $t_w$, in the limit that $t_w \to -\infty$. The bulk dual is a Schwarzschild-AdS black hole with a shock wave on the horizon. If a signal is sent in from the right boundary at $t_R$, there is a latest time that an observer on the left can jump in and receive this signal. This time, $t_L(t_R)$, is a nontrivial function of $t_R$ which depends crucially on the singularity inside the black hole. We have shown how to recover this function from the dual field theory. The hybrid left-right correlator $G(\omega_L, t_R)$ of a high dimension operator has exponential behavior at large imaginary $\omega_L$ with a coefficient that is precisely $t_L(t_R)$.

Much of our analysis applies to all spacetime dimensions $D > 3$, but in Section 4.4 we restricted to $D = 5$ to perform some explicit calculations. We expect the final result will hold in other dimensions.

It is clear that our two-point function does not give information about the singularity for all black holes. First, we need a spacelike singularity to define $t_L(t_R)$, so our calculation cannot be applied to any black hole with an inner horizon. Second, our method requires boundary anchored spacelike geodesics to approach the singularity when their energy becomes large. This will not be true if $g_{tt}$ is bounded from above inside the horizon. (See [32] for examples of black holes with this property.) It is an interesting open question to find boundary observables that probe these other spacelike singularities.

There are a number of additional open questions raised by this work: (1) Our bulk calculation of the correlator corresponds to the large $N$ and large $\lambda$ limit of the dual field theory. It would be interesting to explore quantum or stringy corrections, perhaps along the lines of [33]. (2) More ambitiously, can one compute the correlator (at finite $N$ and $\lambda$) just from the boundary theory? (3) Our analysis (and the earlier analyses [9,10]) depended crucially on having a two-sided black hole. How can one generalize to a single-sided black hole? (4) Suppose that we send in arbitrary neutral, spherical matter from the right. The metric will be time dependent on the right and changed inside the black hole, so our calculation does not apply. However $t_L(t_R)$ is still well defined, and contains the information about how the singularity bends down. Since our final two-point function depends on a right time and left frequency, it is plausible that it will again be dominated by the nearly null spacelike geodesic and have the same exponential behavior with coefficient $t_L(t_R)$. Verifying whether or not this is the case is left for future investigation. A natural first step in this direction would be to try repeating the analysis done in this work in Vaidya geometries.

## Acknowledgments

We would like to thank Steve Shenker for suggesting using bouncing geodesics to detect the singularity bending down. We would also like to thank David Grabovsky, Jesse Held, Robinson Mancilla, and Don Marolf for discussions.

**Funding information** This work was supported in part by NSF Grant PHY-2107939. Y.Z. was supported in part by the National Science Foundation under Grant No. NSF PHY-1748958 and by a grant from the Simons Foundation (815727, LB).

## A On the analytic properties of the frequency space correlator

In this appendix, we will explain several details regarding the definition of our two-point function (38) and its analytic structure. In Section A.1, we start by providing a detailed derivation of $T_{\omega_L,\omega_R}$, the object that allows us to evolve the scalar field modes across the shock wave using (32). As we shall see, its definition involves providing an appropriate $i\epsilon$ prescription. Without the latter we would find our frequency space two-point function to be a distribution on the real axis, which cannot be readily analytically continued. Moving on to Section A.2, we explain the analytic properties of the factors $e^{\pm i\delta_\omega}C(\omega)$ appearing in the frequency space two-point function (38).

### A.1 Derivation of $T_{\omega_L,\omega_R}$

In the main text, in order to evolve the scalar field modes across the shock, we write

$$(-\tilde{V} + \alpha)^{-i\frac{\beta}{2\pi}\omega_R} = \int_{-\infty}^{+\infty} \frac{d\omega_L}{2\pi} T_{\omega_L,\omega_R} e^{-i\omega_L v}. \tag{A.1}$$

By noting that in the past horizon on the left, while staying below the shock, we have $(-\tilde{V} + \alpha)^{-i\frac{\beta}{2\pi}\omega_R} = (e^{\frac{2\pi}{\beta}v} + \alpha)^{-i\frac{\beta}{2\pi}\omega_R}$, it follows that

$$(e^{\frac{2\pi}{\beta}v} + \alpha)^{-i\frac{\beta}{2\pi}\omega_R} = \int_{-\infty}^{+\infty} \frac{d\omega_L}{2\pi} T_{\omega_L,\omega_R} e^{-i\omega_L v}. \tag{A.2}$$

We can then write $T_{\omega_L,\omega_R}$ as the Fourier transform

$$T_{\omega_L,\omega_R} = \int_{-\infty}^{+\infty} dv (e^{\frac{2\pi}{\beta}v} + \alpha)^{-i\frac{\beta}{2\pi}\omega_R} e^{i\omega_L v}. \tag{A.3}$$

One can readily check that this Fourier transform must be a distribution on the real axis because the function we are Fourier transforming is not absolutely integrable. To make this point more explicit, we can compute the indefinite integral

$$
\begin{aligned}
I &= \int dv (e^{\frac{2\pi}{\beta}v} + \alpha)^{-i\frac{\beta}{2\pi}\omega_R} e^{i\omega_L v} \\
&= -\frac{ie^{iv\omega_L}(e^{\frac{2\pi}{\beta}v} + \alpha)^{1-i\frac{\beta}{2\pi}\omega_R}}{\alpha\omega_L} F\left(1, 1 + i\frac{\beta}{2\pi}(\omega_L - \omega_R), 1 + i\frac{\beta}{2\pi}\omega_L, -\frac{e^{\frac{2\pi}{\beta}v}}{\alpha}\right),
\end{aligned}
\tag{A.4}
$$

where $F$ denotes the ordinary hypergeometric function. One can check that $I$ does not converge for $v \to \pm\infty$. However, if we set $\omega_L \to \omega_L - i\epsilon_1$ and $\omega_L - \omega_R \to \omega_L - \omega_R + i\epsilon_2$ with

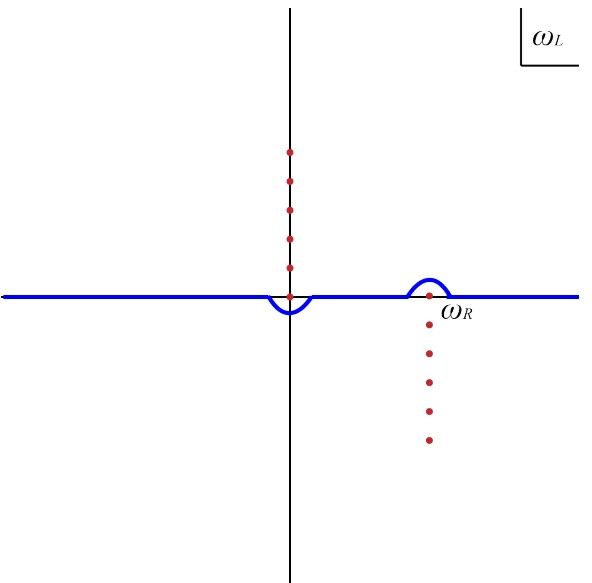

Figure 9: The blue curve is the contour $C$.

$\epsilon_i > 0$, we find

$$\lim_{v \to +\infty} I = \frac{\beta}{2\pi} \frac{\alpha^{i\frac{\beta}{2\pi}(\omega_L - \omega_R + i\epsilon_2)} \Gamma\left(i\frac{\beta}{2\pi}(\omega_L - i\epsilon_1)\right) \Gamma\left(-i\frac{\beta}{2\pi}(\omega_L - \omega_R + i\epsilon_2)\right)}{\Gamma\left(i\frac{\beta}{2\pi}(\omega_R - i\epsilon_1 - i\epsilon_2)\right)}, \tag{A.5}$$

$$\lim_{v \to -\infty} I = 0. \tag{A.6}$$

With this prescription we ensure (A.3) converges and by sending $\epsilon_i \to 0$ at the end we find

$$T_{\omega_L, \omega_R} = \frac{\beta}{2\pi} \frac{\alpha^{i\frac{\beta}{2\pi}(\omega_L - \omega_R)}}{\Gamma(i\frac{\beta}{2\pi}\omega_R)} \Gamma\left(-i\frac{\beta}{2\pi}(\omega_L - \omega_R)\right) \Gamma\left(i\frac{\beta}{2\pi}\omega_L\right). \tag{A.7}$$

While this is the expression we use in the main text, it is important to keep in mind that there is always an implicit $i\epsilon$ prescription associated to it, given by (A.5). In other words, inserting (A.7) in (A.2) does not yield the left-hand side unless we do an appropriate deformation of the contour.

As a consistency check of the above derivation, we write

$$(e^{\frac{2\pi}{\beta}v} + \alpha)^{-i\frac{\beta}{2\pi}\omega_R} = \int_C \frac{d\omega_L}{2\pi} T_{\omega_L, \omega_R} e^{-i\omega_L v}, \tag{A.8}$$

with $T_{\omega_L, \omega_R}$ given by (A.7) and where the contour $C$ is specified below. In the $\omega_L$ plane, $T_{\omega_L, \omega_R}$ has one line of poles given by $\omega_L = \frac{2\pi i n}{\beta}$ and another at $\omega_L = \omega_R - i\frac{2\pi n}{\beta}$, where $n \in \mathbb{N}_0$. The contour $C$ is as shown in Figure 9 and is informed by the $i\epsilon$ prescription in (A.5). We now check that this choice of contour does yield the correct result. We are interested in computing the integral

$$\frac{\beta}{2\pi} \frac{\alpha^{-i\frac{\beta}{2\pi}\omega_R}}{\Gamma(i\frac{\beta}{2\pi}\omega_R)} \int_C \frac{d\omega_L}{2\pi} \Gamma\left(-i\frac{\beta}{2\pi}(\omega_L - \omega_R)\right) \Gamma\left(i\frac{\beta}{2\pi}\omega_L\right) e^{i\left(\frac{\beta}{2\pi}\log\alpha - v\right)\omega_L}. \tag{A.9}$$

If $\frac{\beta}{2\pi}\log\alpha > v$, we close the contour in the upper half plane. The choice of contour $C$ then gives a semi-circle that includes the pole at 0 and excludes the pole at $\omega_R$. This gives the

correct behaviour because when $\alpha \gg 1$ or $v/\beta \ll -1$, the $n = 0$ pole dominates the sum over the residues and one gets $\alpha^{-i\frac{\beta}{2\pi}\omega_R}$. We can compute the integral exactly as follows

$$
\begin{aligned}
&\frac{\beta}{2\pi}\frac{\alpha^{-i\frac{\beta}{2\pi}\omega_R}}{\Gamma\left(i\frac{\beta}{2\pi}\omega_R\right)}\int\frac{d\omega_L}{2\pi}\Gamma\left(-i\frac{\beta}{2\pi}(\omega_L-\omega_R)\right)\Gamma\left(i\frac{\beta}{2\pi}\omega_L\right)e^{i\left(\frac{\beta}{2\pi}\log\alpha-v\right)\omega_L}\\
&=\frac{\beta}{2\pi}\frac{\alpha^{-i\frac{\beta}{2\pi}\omega_R}}{\Gamma\left(i\frac{\beta}{2\pi}\omega_R\right)}i\sum_{n=0,1\ldots}\Gamma\left(n+i\frac{\beta}{2\pi}\omega_R\right)e^{-\left(\log\alpha-\frac{2\pi}{\beta}v\right)n}\text{Res}\left[\Gamma\left(i\frac{\beta}{2\pi}\omega_L\right),\omega_L=\frac{2\pi i n}{\beta}\right]\\
&=\frac{\alpha^{-i\frac{\beta}{2\pi}\omega_R}}{\Gamma\left(i\frac{\beta}{2\pi}\omega_R\right)}\sum_{n=0,1\ldots}\Gamma\left(n+i\frac{\beta}{2\pi}\omega_R\right)e^{-\left(\log\alpha-\frac{2\pi}{\beta}v\right)n}\frac{(-1)^n}{n!}\\
&=(\alpha+e^{\frac{2\pi}{\beta}v})^{-i\frac{\beta}{2\pi}\omega_R},
\end{aligned}
\tag{A.10}
$$

yielding the right result. On the other hand, if $\frac{\beta}{2\pi}\log\alpha < v$, we close the contour in the lower half plane. Now our semi-circle excludes the pole at 0 and includes the pole at $\omega_R$. When $\alpha \ll 1$ or $v/\beta \gg 1$, the $n = 0$ pole dominates the sum and one gets $e^{-i\omega_R v}$. With this, we see that there is a sense in which $T_{\omega_L,\omega_R}$ acts like a delta function $\delta(\omega_L - \omega_R)$ as $\alpha \to 0$, as it should be for consistency of (A.2).[27] Using this contour, we can do an analogous calculation to see that we again get the correct result

$$
\begin{aligned}
&\frac{\beta}{2\pi}\frac{\alpha^{-i\frac{\beta}{2\pi}\omega_R}}{\Gamma(i\frac{\beta}{2\pi}\omega_R)}\int\frac{d\omega_L}{2\pi}\Gamma\left(-i\frac{\beta}{2\pi}(\omega_L-\omega_R)\right)\Gamma\left(i\frac{\beta}{2\pi}\omega_L\right)e^{i\left(\frac{\beta}{2\pi}\log\alpha-v\right)\omega_L}\\
&=-\frac{\beta}{2\pi}\frac{\alpha^{-i\frac{\beta}{2\pi}\omega_R}}{\Gamma(i\frac{\beta}{2\pi}\omega_R)}i\sum_{n=0,1\ldots}\text{Res}\left[\Gamma\left(-i\frac{\beta}{2\pi}(\omega_L-\omega_R)\right)\Gamma\left(i\frac{\beta}{2\pi}\omega_L\right)e^{i\left(\frac{\beta}{2\pi}\log\alpha-v\right)\omega_L},\omega_L=\omega_R-\frac{2\pi i n}{\beta}\right]\\
&=\frac{e^{-iv\omega_R}}{\Gamma(i\frac{\beta}{2\pi}\omega_R)}\sum_{n=0,1\ldots}\Gamma\left(i\frac{\beta}{2\pi}\omega_R+n\right)\frac{(-1)^n}{n!}e^{\left(\log\alpha-\frac{2\pi}{\beta}v\right)n}\\
&=(\alpha+e^{\frac{2\pi}{\beta}v})^{-i\frac{\beta}{2\pi}\omega_R}.
\end{aligned}
\tag{A.13}
$$

## A.2 Analytic structure of $e^{\pm i\delta_\omega}C(\omega)$

In this section, we consider the analytic properties of $e^{\pm i\delta_\omega}C(\omega)$, which directly appears in $G(\omega_L, \omega_R, l)$. It will be useful to introduce solutions to (18) that have different boundary conditions than those satisfied by $\psi_{\omega,l}$. We define $\varphi$ and $f_\pm$ to be solutions with the following properties

$$
\begin{aligned}
\varphi(\omega;z) &\to z^{\frac{1}{2}+v}, \quad z \to 0,\\
f_\pm(\omega;z) &\to e^{\pm i\omega z}, \quad z \to \infty,
\end{aligned}
\tag{A.14}
$$

---

[27]Rigorously speaking, to define the $T_{\omega_L,\omega_R}$ yielded by (A.3) in the real axis, one should keep the $i\epsilon_i$ factors in (A.5) and take $\epsilon_i \to 0$ with more care than what we did when writing (A.7). In particular, preserving the $i\epsilon$ prescription, we can write

$$
T_{\omega_L,\omega_R} = \frac{f(\omega_L,\omega_R)}{(\omega_L - i\epsilon_1)} + \frac{g(\omega_L,\omega_R)}{(\omega_L - \omega_R + i\epsilon_2)},
\tag{A.11}
$$

where $f$ and $g$ are defined and continuous for every $\omega_{L/R} \in \mathbb{R}$. Invoking the fact that under an integral we have in the distributional sense

$$
\lim_{\epsilon\to 0^+}\frac{1}{x\pm i\epsilon} = \mp i\pi\delta(x) + \mathcal{P}\left(\frac{1}{x}\right),
\tag{A.12}
$$

where $\mathcal{P}$ denotes the Cauchy principal value, the limit $\epsilon_i \to 0^+$ of (A.11) gives rise to the actual distribution yielded by (A.3), which contains a factor with $\delta(\omega_L - \omega_R)$. That should be the only surviving contribution when $\alpha \to 0$ for consistency of (A.2).

i.e. $\varphi$ is the normalizable mode, $f_+$ ($f_-$) is the pure ingoing (outgoing) mode at the horizon. The advantage of introducing these solutions is that their analyticity properties are easily deduced from standard methods in scattering theory [34]: $\varphi(\omega; z)$ is an entire function of $\omega$ and $f_+(\omega; z)$ is analytic in the upper half $\omega$ plane. Viewed as complex functions of $\omega$, these solutions have the following reflection properties.

$$f_+(\omega; z)^* = f_-(\omega^*; z) = f_+(-\omega^*; z),$$
$$\varphi(\omega; z)^* = \varphi(\omega^*; z), \quad \varphi(\omega; z) = \varphi(-\omega; z). \tag{A.15}$$

In particular, the properties of $f_-(\omega; z)$ can be easily deduced from those of $f_+(\omega; z)$ by reflection.

Using the specific properties of (18) we can constrain the singularities of $f_+(\omega; z)$ in the lower half $\omega$ plane. Since $r$ is a periodic function of $z$ with period $i\beta/2$ for $Re(z) \gg 1$, the potential $V(z)$ can be expanded as a sum of exponentials at large real $z$

$$V(z) = \sum_{n=1}^{\infty} a_n e^{-\frac{4\pi n}{\beta} z}. \tag{A.16}$$

This implies that the only singularities of $f_+(\omega; z)$ in the lower half $\omega$ plane are equally spaced simple poles along the imaginary axis at

$$\omega = -i\frac{2\pi n}{\beta}, \quad n = 1, 2, \dots \tag{A.17}$$

The Jost function $\mathcal{F}(\omega)$ is defined as the linear coefficients when $\varphi(\omega; z)$ is written in terms of a linear combination of $f_\pm(\omega; z)$

$$\varphi(\omega, z) = \frac{1}{2i\omega} \left( \mathcal{F}(-\omega) f_+(\omega, z) - \mathcal{F}(\omega) f_-(\omega, z) \right). \tag{A.18}$$

The properties of $\mathcal{F}(\omega)$ can be deduced from those of $\varphi(\omega; z)$ and $f_+(\omega; z)$ since $\mathcal{F}(\omega)$ is simply the Wronskian between them

$$\mathcal{F}(\omega) = f_+(\omega; z)\partial_z \varphi(\omega; z) - \varphi(\omega; z)\partial_z f_+(\omega; z). \tag{A.19}$$

This implies $\mathcal{F}(\omega)$ must have exactly the same analyticity properties as $f_+(\omega)$, i.e. the only singularities are a line of poles at (A.17). (A.15) implies the reflection property $\mathcal{F}(\omega)^* = \mathcal{F}(-\omega^*)$. $\mathcal{F}(\omega)$ also have the property that its zeroes in the upper half plane correspond to bound states of the potential $V(z)$. Since there are no bound states at $l = 0$, $\mathcal{F}(\omega)$ has no zeroes in the upper half plane. Moreover, note that since $\mathcal{F}(\omega)$ is the coefficient of the outgoing mode in the normalizable mode $\phi$ in (A.18), the zeroes of $\mathcal{F}(\omega)$ in the lower half plane corresponds to the quasinormal modes of the scalar field. The reflection property of $\mathcal{F}(\omega)$ then implies that these quasinormal modes have a reflection symmetry about the imaginary axis.

In terms of these functions, our original modes $\psi_\omega(z)$ can be expressed as

$$\psi_\omega(z) = e^{i\delta_\omega} f_+(\omega; z) + e^{-i\delta_\omega} f_-(\omega; z) = C(\omega)\varphi(\omega; z). \tag{A.20}$$

Comparing with (A.18) gives

$$e^{i\delta_\omega} C(\omega) = -\frac{2i\omega}{\mathcal{F}(\omega)}, \quad e^{-i\delta_\omega} C(\omega) = \frac{2i\omega}{\mathcal{F}(-\omega)}. \tag{A.21}$$

Using the analyticity properties of $\mathcal{F}(\omega)$, we obtain the following properties of $e^{i\delta_\omega} C(\omega)$.

- It is analytic in the upper half plane and its singularities in the lower half plane correspond to the zeroes of $\mathcal{F}(\omega)$, which as mentioned are the quasinormal modes.

- It has a line of zeroes along the negative imaginary axis $\omega = -i\frac{2\pi n}{\beta}$, $n = 0, 1, \ldots$ Note the difference compared to (A.17) due to $\omega$ in the numerator.

- It has a reflection symmetry about the imaginary axis $e^{i\delta_{-\omega^*}} C(-\omega^*) = (e^{i\delta_\omega} C(\omega))^*$.

Similar statements can be made about $e^{-i\delta_\omega} C(\omega)$ since it is simply related to $e^{i\delta_\omega} C(\omega)$ via a reflection about the real axis $e^{i\delta_{\omega^*}} C(\omega^*) = (e^{-i\delta_\omega} C(\omega))^*$.

## B    An extension of the thermal product formula

In this appendix we provide a self-contained derivation of the results summarized in 4.1.2. While we made some restrictions in 4.1.2, we derive the result in full generality in this appendix. What follows is expected to be valid for any reasonable asymptotically AdS black hole of the form (2) with a shock wave at the horizon. The dependence on angular modes will be kept implicit because the derivation does not depend on them explicitly. We will however assume there are no imaginary quasinormal modes throughout the derivation - as we explain at the end it is simple to include them.

We start by presenting the result derived in [21], which is the main inspiration for this derivation. The holographic thermal two-point function can be written as

$$G_{\text{thermal}}(\omega) = \frac{G_{\text{thermal}}(0)}{\prod_{n=1}^{\infty} \left(1 - \frac{\omega^2}{\omega_n^2}\right)\left(1 - \frac{\omega^2}{(\omega_n^*)^2}\right)}, \tag{B.1}$$

where $\omega_n$, $-\omega_n^*$ correspond to the quasinormal modes. Moving on to our own two-point function computed in (38), we note that it admits the decomposition

$$G(\omega_L, \omega_R) = \frac{\nu^2 \beta^2}{(2\pi)^2 \pi} \Gamma\left(\frac{\beta}{2\pi} i\Delta\omega\right) \alpha^{-i\frac{\beta}{2\pi}\Delta\omega} G_L(\omega_L) G_R(\omega_R), \tag{B.2}$$

where we defined

$$G_L(\omega_L) = \Gamma\left(-\frac{\beta}{2\pi} i\omega_L\right) e^{i\delta_{\omega_L}} C(\omega_L) = -\Gamma\left(-\frac{\beta}{2\pi} i\omega_L\right) \frac{2i\omega_L}{\mathcal{F}(\omega_L)}, \tag{B.3}$$

$$G_R(\omega_R) = \Gamma\left(\frac{\beta}{2\pi} i\omega_R\right) e^{-i\delta_{\omega_R}} C(\omega_R) = \Gamma\left(\frac{\beta}{2\pi} i\omega_R\right) \frac{2i\omega_R}{\mathcal{F}(-\omega_R)}, \tag{B.4}$$

with $\Delta\omega = \omega_L - \omega_R$ and used (A.21) to write the second equality. One can check that the above functions satisfy the properties $G_{L/R}(\omega)^* = G_{L/R}(-\omega^*)$ and $G_L(\omega) = G_R(-\omega)$, where the former follows from the fact that $\mathcal{F}(\omega)^* = \mathcal{F}(-\omega^*)$ as explained in Appendix A.2. It also follows from the results in Appendix A.2 that both functions are meromorphic and their reciprocal $1/G_{L/R}(\omega)$ are entire. The Hadamard factorization theorem [35] states that an entire function $f(z)$ of order $m$[28] with roots $a_n \neq 0$ can be decomposed as

$$f(z) = z^l e^{Q(z)} \prod_{n=1}^{\infty} E_{\lfloor m \rfloor}(z/a_n), \tag{B.5}$$

where $Q(z)$ is a polynomial of degree $q \leq m$, $\lfloor m \rfloor$ denotes the integer part of $m$, $l$ corresponds to the order of the zero of $f(z)$ at $z = 0$ (with the understanding that $l = 0$ if there is no zero)

---

[28]An entire function $f(z)$ is said to be of finite order if there exist $a, r > 0$ such that

$$\left|f(z)\right| \leq e^{|z|^a},$$

for all $|z| > r$. The infimum of all such $a$ is what we call the order of the function.

and

$$E_{\lfloor m \rfloor}(z) = (1-z) \prod_{k=1}^{\lfloor m \rfloor} e^{z^k/k}. \tag{B.6}$$

Since $1/G_{L/R}(\omega)$ are entire and we know their roots, we just need to determine their order $m$ to apply the theorem. The rigorous definition of order for an entire function is given in footnote 28, but, roughly speaking, if the function at hand behaves as $e^{z^b}$ asymptotically, the order of the function is $b$. Using the results at large $\omega$ in [29] and our own WKB analysis, we expect generically that $1/G_{L/R}(\omega)$ are entire functions of order $m = 1$. We can show this explicitly for large operator dimensions in $D = 5$ Schwarzschild-AdS by using the results in [29] or looking at our own expression (100). In passing, we note that this implies that $\alpha^{-i\frac{\beta}{2\pi}\Delta\omega}/G(\omega_L, \omega_R)$ itself is also an entire function of order 1 because the $\Gamma$ in (B.2) does not increase the order of the function.

We can now apply the Hadamard factorization theorem described above to obtain

$$\frac{1}{G_L(\omega)} = e^{c_0 + c_1 \omega} \prod_{n=1}^{\infty} \left(1 - \frac{\omega}{\omega_n}\right)\left(1 + \frac{\omega}{\omega_n^*}\right) e^{\omega/\omega_n} e^{-\omega/\omega_n^*}, \tag{B.7}$$

where $(\omega_n, -\omega_n^*)$ denote the poles of $G_L(\omega)$ and $c_0$ and $c_1$ are arbitrary, possibly complex, constants. We note that in this case the poles are exactly the quasinormal modes. Due to the reflection property $G_L(\omega)^* = G_L(-\omega^*)$, we must have

$$\frac{G_L(\omega)^*}{G_L(-\omega^*)} = e^{c_0 - c_0^*} e^{-\omega^*(c_1 + c_1^*)} = 1, \tag{B.8}$$

where we used the fact that our set of poles $\mathcal{S} = \{\omega_n, -\omega_n^*\}$ has the property $\mathcal{S}^* = -\mathcal{S}$ to cancel out the factors inside the infinite products. This implies we must have $c_0 = c_0^*$ and $c_1 = -c_1^*$, or in other words, $c_0$ must be a real constant and $c_1$ purely imaginary. This derivation readily extends to $G_R(\omega)$ since its set of poles $\{-\omega_n, \omega_n^*\}$ is simply the reflection of the quasinormal modes with respect to the real axis. In fact, noting that $G_L(\omega) = G_R(-\omega)$, we readily have

$$\frac{1}{G_R(\omega)} = e^{c_0 - c_1 \omega} \prod_{n=1}^{\infty} \left(1 + \frac{\omega}{\omega_n}\right)\left(1 - \frac{\omega}{\omega_n^*}\right) e^{-\omega/\omega_n} e^{\omega/\omega_n^*}. \tag{B.9}$$

While $c_0$ and $c_1$ are undetermined constants, they have a simple relationship with the functions $G_{L/R}(\omega)$, namely

$$e^{-c_0} = G_L(0) = G_R(0), \tag{B.10}$$

$$c_1 = -\frac{G_L'(0)}{G_L(0)} = \frac{G_R'(0)}{G_R(0)}. \tag{B.11}$$

Using the results in [10] for a two-sided thermal two-point function, we note that

$$G_L(\omega) G_R(\omega) = \frac{2\pi^2}{\beta v^2} G_{\text{thermal}}(\omega), \tag{B.12}$$

and we can write our two-point function (B.2) as

$$G(\omega_L, \omega_R) = \frac{G_\Delta(\Delta\omega) G_{\text{thermal}}(0)}{\Pi(\omega_L)\Pi(-\omega_R)}, \tag{B.13}$$

where we defined

$$\Pi(\omega) = \prod_{n=1}^{\infty}\left(1 - \frac{\omega}{\omega_n}\right)\left(1 + \frac{\omega}{\omega_n^*}\right), \tag{B.14}$$

$$G_\Delta(\Delta\omega) = \frac{\beta}{2\pi}\Gamma\left(\frac{\beta}{2\pi}i\Delta\omega\right)\alpha^{-i\frac{\beta}{2\pi}\Delta\omega}e^{-c_1\Delta\omega}\prod_{n=1}^{\infty}e^{-\Delta\omega/\omega_n}e^{\Delta\omega/\omega_n^*}. \tag{B.15}$$

Since the $\Gamma$ function admits a known Hadamard factorization, we can write $G_\Delta(\Delta\omega)$ as

$$G_\Delta(\Delta\omega) = \frac{e^{ig_\alpha(\Delta\omega)}}{\Delta\omega\Pi_\Delta(\Delta\omega)}, \tag{B.16}$$

where

$$g_\alpha(\Delta\omega) = \frac{\beta}{2\pi}\Delta\omega\,(c - \gamma - \log\alpha) - \Delta\omega\sum_{n=1}^{\infty}\left(2\,\mathrm{Im}\left\{\frac{1}{\omega_n}\right\} - \frac{1}{\Omega_n}\right) - \frac{\pi}{2}, \tag{B.17}$$

$$\Pi_\Delta(\Delta\omega) = \prod_{n=1}^{\infty}\left(1 - \frac{\Delta\omega}{i\Omega_n}\right). \tag{B.18}$$

$\Omega_n = 2\pi n/\beta$ are Matsubara frequencies, we defined $c = i2\pi c_1/\beta$, and $\gamma$ is the Euler's constant. Summing it up, we can write our two-point function in full closed form as a product over quasinormal modes and Matsubara frequencies up to a rescaling and a phase[29]

$$G(\omega_L, \omega_R) = \frac{G_{\mathrm{thermal}}(0)e^{ig_\alpha(\Delta\omega)}}{\Delta\omega\Pi(\omega_L)\Pi(-\omega_R)\Pi_\Delta(\Delta\omega)}, \tag{B.19}$$

with

$$\Delta\omega = \omega_L - \omega_R. \tag{B.20}$$

One can check that the thermal two-point function (B.1) is captured by the residue at $\Delta\omega = 0$ of (B.19), namely

$$\mathrm{Res}_{\Delta\omega=0}\,G(\omega_L, \omega_R) = -iG_{\mathrm{thermal}}(\omega), \tag{B.21}$$

where we set $\omega_L = \omega_R = \omega$. We explain the physics underlying this fact in the main text. Writing the poles explicitly, we have

$$G(\omega_L, \omega_R) = \frac{G_{\mathrm{thermal}}(0)e^{ig_\alpha(\Delta\omega)}}{\Delta\omega\prod_{n=1}^{\infty}\left(1 - \frac{\Delta\omega}{i\Omega_n}\right)\left(1 - \frac{\omega_L}{\omega_n}\right)\left(1 + \frac{\omega_L}{\omega_n^*}\right)\left(1 + \frac{\omega_R}{\omega_n}\right)\left(1 - \frac{\omega_R}{\omega_n^*}\right)}, \tag{B.22}$$

with

$$g_\alpha(\Delta\omega) = \frac{\beta}{2\pi}\Delta\omega\,(c - \gamma - \log\alpha) - \Delta\omega\sum_{n=1}^{\infty}\left(2\,\mathrm{Im}\left\{\frac{1}{\omega_n}\right\} - \frac{1}{\Omega_n}\right) - \frac{\pi}{2}, \tag{B.23}$$

which is the result we present in the main text in eqs. (46) and (47).

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
