# Peer review of "Boundary signature of singularity in the presence of a shock wave"

_SciPost Physics, doi:SciPost Phys. 16, 060 (2024)_

## Round 1 · Referee Report · Anonymous (Referee 1) · 2024-1-9

Report

The paper focuses on studying the imprints of bulk curvature singularities onto boundary observables. Previous literature has always focused on analytic solutions, making it unclear whether the imprints one gets come from proper black hole dynamics or are a result of analyticity. In this respect, the authors study shockwave perturbations of the Schwarzschild-AdS black hole with the goal of seeing whether the boundary observables still encode some information about the bulk curvature singularity.

The shockwave breaks the boost symmetry in the bulk, so the method that the authors use is slightly different than the ones from previous literature; in fact, the authors combine the previous methods and study real-time correlation functions at a large and imaginary frequency (necessary for ''reaching'' the singularity). In other words, since there is no boost symmetry, the full correlator will depend on two variables - left and right times (or left and right frequencies). The goal of the authors is to keep one of the times and Fourier-transform the other - this way, they can obtain an easier imprint (via real geodesics, as in the Festuccia-Liu paper) of the singularity while keeping track of one of the infall times, matching it with the pure bulk calculation.

Along the way, they also extend a previously found exact and beautiful formula for the thermal 2-sided 2-point function: the authors note that the same Hadamard factorization property holds for the shockwave case, and they manage to express the frequency correlator as a product over quasinormal modes and Matsubara frequencies.

The paper studies a well-defined question of importance, testing if the previously found methods give trivial results that stem from analyticity or if they truly encode some information about the singularity. Their paper indicates that the methods do capture information about the singularity even in the presence of a shockwave. The paper is nicely written, and I can recommend the paper to be published.

Small question:

In Sec. 2, you briefly recall the methods from Fidkowski et al. and Festuccia-Liu. You mention that in the first paper, the bouncing geodesics were not the dominant ones for the boundary correlator, yet the second method allows us to rely only on such bouncing geodesics instead of reading off the subtle signal from another sheet as in the first method. Do you understand why such a difference emerges (that is, how come one method is much more subtle than the other), and can you elaborate on your understanding?

  • validity: top
  • significance: high
  • originality: top
  • clarity: top
  • formatting: perfect
  • grammar: perfect

Author:  Leonel Queimada  on 2024-01-30  [id 4293]

(in reply to Report 1 on 2024-01-09)

We would like to thank the referee for their comments and question. An answer to the question can be found below:

The main reason why the method of Festuccia-Liu allows us to "see" the bouncing geodesic more directly is due to a direct connection between the correlator along the imaginary axis of frequency space and real bulk geodesics. This can be established through a WKB approximation. When staying in position space, the connection between real bulk geodesics and the correlator is less clear. While we expect the method of Festuccia-Liu to generically capture real bulk geodesics in the way we described, we know that in position space the relationship between real geodesics and the correlator varies depending on the spacetime we consider. For example, the results in Fidkowski et al. suggest that in Schwarzschild-AdS d3 we need to consider an analytic continuation to a second sheet. On the other hand, we know that in BTZ the two-sided correlator is simply given by real bulk geodesic lengths in the semiclassical limit.

---

## Round 1 · Referee Report · Anonymous (Referee 2) · 2024-1-12

Report

In this paper, the authors examine the effect of a shockwave on the thermal two-point function, and explain how to diagnose the presence of the shockwave via bouncing geodesics. This is an inventive and interesting paper and I recommend it for publication.

Two questions:

1. The authors choose to work with a hybrid correlator with one frequency variable and one time variable. It seems that an alternate strategy is to work with only time variables, but with a nonstandard choice of imaginary frequency contour, as in Figure 10 of hep-th/0506202. If one chooses this contour, does the correlator G(t_L,t_R) have a singularity corresponding to bouncing geodesics? If so it may be worth commenting on this.

2. This paper derives an interesting generalization of the thermal product formula. In hep-th/2304.12339 the OPE limit along with the product formula were used to derive asymptotic constraints on QNMs. Have the authors thought about whether a similar analysis can be applied for their shockwave correlator to derive additional constraints on QNMs?

  • validity: top
  • significance: high
  • originality: high
  • clarity: top
  • formatting: perfect
  • grammar: perfect

Author:  Leonel Queimada  on 2024-01-30  [id 4292]

(in reply to Report 2 on 2024-01-12)
Category:
answer to question

We would like to thank the referee for their comments and questions. Answers to the questions can be found below:

  1. If we understand it correctly, the referee refers to the object H12(τ) defined in Figure 10 of hep-th/0506202 by integrating along the imaginary axis in frequency space. We have considered generalizing that object to our case but there are some problems associated to doing so. More specifically, in our case there can be lines of Matsubara poles along the imaginary axis which render the choice of the contour C2 in hep-th/0506202 badly defined. A similar integral might be defined by deforming the contour around the poles, but there is no unambiguous way to do so. Furthermore, such an integral would be dominated by the saddle in our eq. 4.90, which is different from the saddle picked up by our current setup (eq. 4.89). It is unclear what physical information is captured by this saddle. For these reasons, we decided to not consider it in our paper.

  2. This is a very interesting question for which we do not have an immediate answer. It would be good to consider this question in more detail in future work. In hep-th/2304.12339 a thermal correlator is studied. For that reason, the authors possess results regarding the OPE which they can use to constrain the QNMs. We are studying a two-point function in a shockwave background which relates to the structure of thermal four-point functions. We believe that in order to answer this question carefully, a detailed study of thermal four-point functions in frequency space is necessary. In particular, one would have to understand the situations/regimes in which holographic thermal four-point functions admit this kind of decompositions.

---

## Round 2 · Referee Report · Anonymous (Referee 1) · 2024-2-6

Report

The authors have replied to my question, therefore, I recommend the paper be published.

---

## Round 2 · Referee Report · Anonymous (Referee 2) · 2024-2-6

Report

The authors have addressed my questions, I recommend this paper for publication.

---

## Round 2 · Author Response

We are resubmitting a new version of the manuscript with added references. We added the replies to the referees' questions below their reports.

---

## Round 2 · List of Changes

- Added references.

---

## Editorial Decision

published